# SENP6 regulates localization and nuclear condensation of DNA damage response proteins by group deSUMOylation

Laura A. Claessens[1], Matty Verlaan-de Vries[1], Ilona J. de Graaf[1] & Alfred C. O. Vertegaal [1]✉

The SUMO protease SENP6 maintains genomic stability, but mechanistic understanding of this process remains limited. We find that SENP6 deconjugates SUMO2/3 polymers on a group of DNA damage response proteins, including BRCA1-BARD1, 53BP1, BLM and ERCC1-XPF. SENP6 maintains these proteins in a hypo-SUMOylated state under unstressed conditions and counteracts their polySUMOylation after hydroxyurea-induced stress. Co-depletion of RNF4 leads to a further increase in SUMOylation of BRCA1, BARD1 and BLM, suggesting that SENP6 antagonizes targeting of these proteins by RNF4. Functionally, depletion of SENP6 results in uncoordinated recruitment and persistence of SUMO2/3 at UVA laser and ionizing radiation induced DNA damage sites. Additionally, SUMO2/3 and DNA damage response proteins accumulate in nuclear bodies, in a PML-independent manner driven by multivalent SUMO-SIM interactions. These data illustrate coordinated regulation of SUMOylated DNA damage response proteins by SENP6, governing their timely localization at DNA damage sites and nuclear condensation state.

In order to maintain genome integrity, cells are constantly sensing and repairing DNA damage induced by both exogenous and endogenous sources through an integrated network of signaling pathways and repair mechanisms, collectively termed the DNA damage response (DDR). The DDR is tightly regulated by post-translational modifications of proteins, including phosphorylation, acetylation, PARylation, ubiquitination, and modification by small ubiquitin-like modifier protein (SUMO)[1]. Proteomics has enabled the identification of thousands of SUMOylated proteins, including proteins involved in the DDR[2]. Indeed, many DDR factors become SUMOylated in response to DNA damage and SUMO aids their accumulation at local sites of DNA damage[3-6].

SUMO can be conjugated to a single lysine residue (mono-SUMOylation) or multiple lysine residues of target proteins (multi-SUMOylation). Like ubiquitin, two functional SUMO isoforms in mammalian cells, SUMO2 and −3 (referred to as SUMO2/3) are capable of forming polymers via internal SUMOylation consensus motifs (polySUMOylation)[7,8]. SUMOylation enables protein-protein interaction and protein complex formation through non-covalent interactions with SUMO-interacting motifs (SIMs)[9]. By protein-group modification, SUMOylation can affect the activity, stability or localization of multiple functionally or physically related proteins within a cellular pathway, which can explain the redundancy of individual SUMO modification events[10]. Group modification by SUMO is particularly important for DNA repair in yeast[11].

Extensive crosstalk exists between SUMOylation and ubiquitination during the DDR[12]. Multi- or polySUMOylated proteins can be recognized by SUMO-targeted ubiquitin ligases (STUbLs). STUbLs ubiquitinate these proteins and promote their degradation by the proteasome, thereby enabling timely protein turnover during the DDR. The best-characterized mammalian STUbLs include RNF4 and RNF111[13]. Many DDR factors are directly or indirectly regulated by RNF4, including BRCA1-BARD1, MDC1, RPA70 and the Fanconi Anemia ID complex (FANCI and FANDC2)[14-18].

SUMOylation and ubiquitination are both highly dynamic and reversible processes, counterbalanced by the activity of deconjugating enzymes. Deubiquitinating enzymes play an important role in multiple

---

[1]Cell and Chemical Biology, Leiden University Medical Centre, Leiden, The Netherlands. ✉e-mail: vertegaal@lumc.nl

DNA repair pathways, including USP7 which deubiquitinates and stabilizes RNF168 at double-strand breaks[19]. However, we are currently limited in our understanding of SUMO deconjugation in the DDR. SUMO deconjugation is mainly achieved by SUMO-specific proteases of the SENP family, SENP1-3 and SENP5-7[20]. SENP2 was found to promote DNA repair through preventing excessive SUMOylation of MDC1 and its clearance by RNF4 from double-strand breaks[21]. Moreover, SENP7, a protease responsible for the deconjugation of SUMO2/3 polymers on proteins, was found to promote chromatin relaxation during homologous recombination through the deconjugation of KAP1[22]. Recently, we found by proteomics that SENP6, another protease capable of deconjugating SUMO2/3 polymers, regulates a group of DDR factors of which many are involved in double-strand break repair[23]. Similarly, in another recent proteomics screen, SENP6 was identified as a key regulator of sister chromatid cohesion and chromatin residency of proteins involved in the ATR-Chk1 DNA damage checkpoint[24]. In addition, SENP6 was described to counteract SUMOylation of the DDR factors EXO1, RPA70, and the Fanconi Anemia ID complex[16,17,25].

Here, we sought to mechanistically delineate how SENP6 regulates the DDR. We confirmed that SENP6 counteracts polySUMOylation of a large functional group of DDR factors, including the BRCA1-BARD1 heterodimer, 53BP1, BLM, and the ERCC1-XPF endonuclease. SENP6 maintains these proteins in a hypoSUMOylated state under unstressed conditions as well as after hydroxyurea-induced genotoxic stress, during active DNA damage signaling. Co-depletion of SENP6 and RNF4 led to a further increase in SUMOylation of BRCA1, BARD1, and BLM compared to SENP6 depletion alone, suggesting that one function of SENP6 is to antagonize targeting of these proteins by the RNF4-STUbL pathway. Functionally, depletion of SENP6 resulted in uncoordinated recruitment and persistence of SUMO2/3 at UVA laser micro-irradiation and ionizing radiation (IR) induced DNA damage sites. Additionally, SUMO2/3 and DDR proteins accumulated in nuclear bodies, in a PML-independent manner driven by multivalent SUMO-SIM interactions.

## Results

### SENP6 is required for maintaining genomic stability

SENP6 is critical for cell cycle progression and cell survival[17,23,26]. Moreover, the knockdown of SENP6 results in an increase in γH2AX foci and an overall increase in γH2AX levels, demonstrating that SENP6 is required for genome stability[17,24,27]. We confirmed a similar increase in the number of γH2AX foci upon knockdown of SENP6, as well as the number of SUMO2/3 foci, as determined by immunofluorescence (Fig. 1a–c). This was reversed by treating with the SUMO-E1 inhibitor ML792[28], indicating that the DNA damage observed in the absence of SENP6 is induced through excessive SUMOylation (Fig. 1c, Supplementary Fig. 1a). Efficient knockdown of SENP6 and inhibition of SUMO conjugation were confirmed by immunoblotting (Supplementary Fig. 1b). SUMO localizes to sites of DNA damage[29,30] and SUMO2/3 was previously shown to colocalize with γH2AX foci induced by SENP6 knockdown[17]. Indeed, we observed a significant increase in SUMO2/3 and γH2AX colocalization in foci in the absence of SENP6 (Fig. 1d). Knockdown of SENP6 also led to a minor increase in SUMO1 foci, and SUMO1 and γH2AX colocalization, although to a much lesser extent (Supplementary Fig. 1c). The increase in γH2AX foci was specific for SENP6, as knockdown of SENP7, the other polySUMO2/3 specific deconjugase, did not result in an increase in the number of γH2AX foci (Supplementary Fig. 1d). Further pointing towards genomic instability was our previous observation that knockdown of SENP6 led to an increase in the number of micronuclei[23], which we again confirmed here (Fig. 1e). This is indicative of defective chromosome segregation during mitosis, which can be a result of lagging whole chromosomes or chromosome fragments generated through DNA double-strand breaks. Indeed, the majority of micronuclei were γH2AX⁺ (Fig. 1e). Since RAD51 depletion has commonly been observed as off-target

effect in RNAi screens and is a critical protein in the DDR, we ruled out that knockdown of SENP6 with the siRNA pool used in this study affected RAD51 protein levels (Supplementary Fig. 1e).

Both DNA repair by homologous recombination and non-homologous end-joining were previously found to require SENP6[22]. The occurrence of spontaneous DNA damage, together with a decreased efficiency in the repair of induced DNA damage, indicates that cells are more sensitive to DNA damage-inducing events in the absence of SENP6. We evaluated the sensitivity of SENP6-deficient cells for DNA damage induced by IR, camptothecin (CPT, a topoisomerase I inhibitor), mitomycin C (MMC, a DNA cross-linking reagent), and hydroxyurea (HU, a ribonucleotide reductase inhibitor). For all treatments, we observed reduced cell viability in the absence of SENP6 five days after the start of treatment (Fig. 1f). Collectively, these data illustrate a need for SENP6 in maintaining genomic integrity.

### SENP6 cannot be readily observed at sites of DNA damage

SUMO proteases of the SENP family differ in their subcellular localization[20]. SENP6 localizes to the nucleoplasm and shows diffuse nucleoplasmic staining under control conditions[23,26]. In cellular fractionation assays, SENP6 was found in both the chromatin and nucleoplasmic fraction of untreated and aphidicolin-treated cells[24]. A catalytic dead variant of HA-tagged SENP6 was found to localize to FANCI foci in response to MMC treatment, but not in response to laser micro-irradiation, in a transient overexpression system[16]. We first investigated whether endogenous SENP6 localizes to and accumulates at DNA double-strand breaks after IR. Endogenous SENP6 did not colocalize with γH2AX as determined by immunofluorescence (Supplementary Fig. 1f, Supplementary Fig. 2). We then evaluated the localization of GFP-tagged wildtype SENP6 (GFP-SENP6$_{WT}$) and catalytic dead SENP6 (GFP-SENP6$_{CD}$) after IR. Both GFP-SENP6$_{WT}$ and GFP-SENP6$_{CD}$ did not colocalize with γH2AX (Supplementary Fig. 1f, Supplementary Fig. 2). The expression of SENP6 appeared diffuse throughout the nucleoplasm under all conditions and did not accumulate in any areas. Adding a pre-extraction step to the protocol in an attempt to visualize only chromatin-bound SENP6, showed the same localization (Supplementary Fig. 2). We then investigated whether SENP6 localizes and accumulates at DNA damage tracks induced by UVA laser micro-irradiation. Exogenously expressed GFP-SENP6$_{WT}$ and GFP-SENP6$_{CD}$ could rarely be observed at the DNA damage tracks, in 12% and 18% of the cells, respectively (Fig. 1g). The endogenous protein was never observed at the DNA damage tracks.

### SUMO2/3 polymers accumulate on DDR proteins in absence of SENP6

SENP6 is a SUMO protease with a preference for deconjugating SUMO2/3 polymers. Using our SUMO2 purification methodology[31] combined with knockdown of SENP6 and label-free quantitative proteomics, we previously identified 180 potential SENP6 substrates, including key players in the DDR[23]. In this approach, U2OS cells stably expressing His10-tagged SUMO2 were treated with lentivirus encoding either a nontargeting shRNA or one of two unique shRNAs targeting SENP6. SUMO2 conjugates were purified by means of His10-pulldown and identified by mass-spectrometry (Fig. 2a). Gene ontology enrichment analysis on the 180 SENP6 substrates revealed that SENP6 regulates DDR processes (Supplementary Fig. 3). Here, we confirmed that knockdown of SENP6 led to an increase of high-molecular-weight SUMO2/3 conjugates and a decrease in free, unconjugated SUMO2/3 (Fig. 2b)[23]. We confirmed that the knockdown of SENP6 predominantly resulted in an increase in total SUMO2/3 conjugates rather than SUMO1 conjugates, as described previously (Supplementary Fig. 4a)[26,27]. In alignment with our proteomics data, we confirmed build-up of SUMO2/3 polymers on the following proteins involved in maintaining genome stability: RAP80, EME1, MUS81, XPF, 53BP1, BLM, CtIP, ERCC1, BARD1, BRCA1 and MDC1 (Fig. 2c).

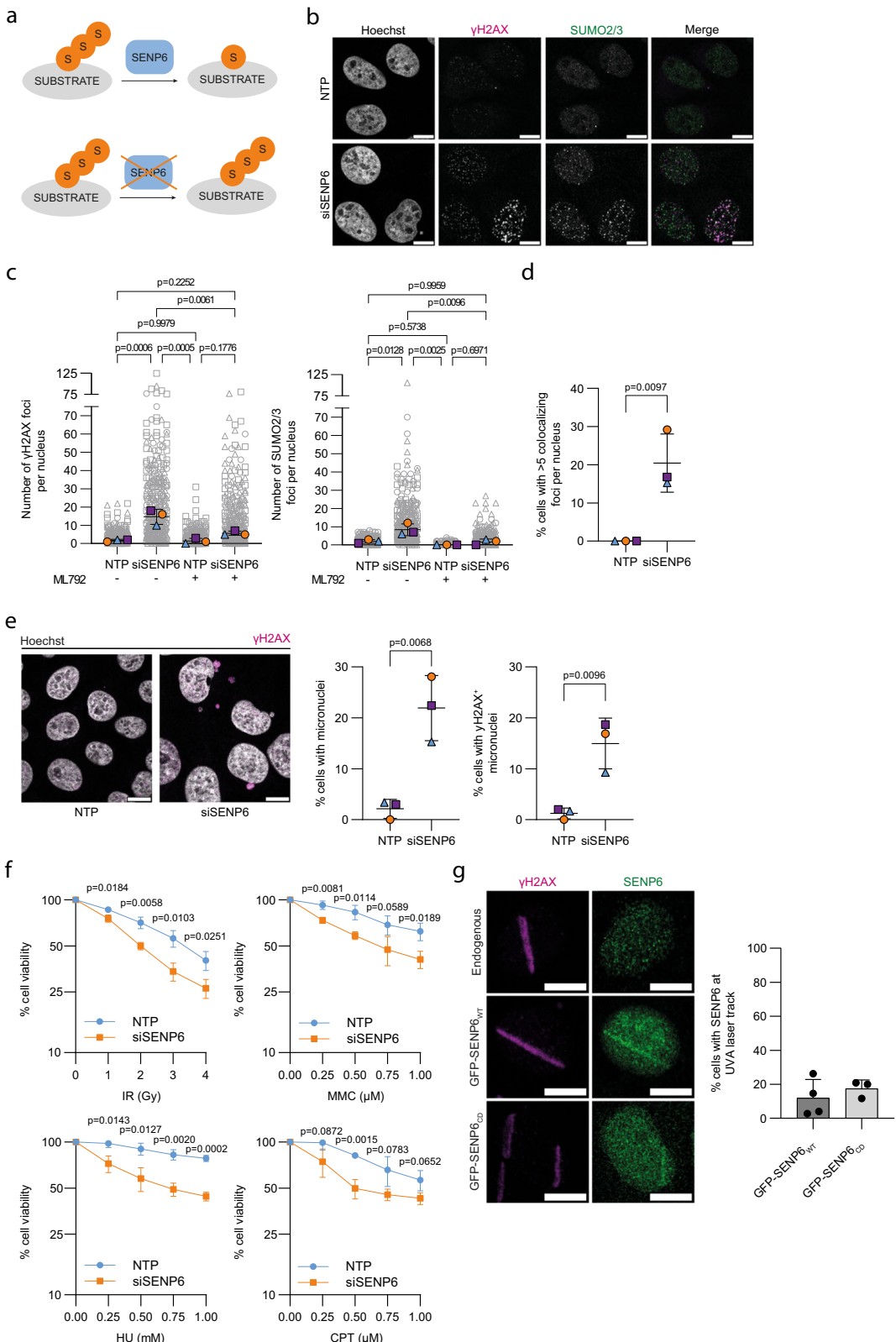

Since the increase in high-molecular-weight SUMO2/3 conjugates in the absence of SENP6 could also be a result of proteins becoming extensively multi-monoSUMOylated rather than poly-SUMOylated, we performed the same His10-pulldown experiment in both U2OS cells stably expressing His10-SUMO2 wildtype and lysine-deficient His10-SUMO2 (His10-SUMO2$_{KO}$), preventing the build-up of His10-SUMO2 chains in the latter. We performed immunoblotting against a selection of the identified SENP6 targets and predominantly observed polySUMOylation of these proteins in the absence of SENP6 rather than multi-monoSUMOylation (Fig. 3a, b). This suggests that SENP6 mainly counteracts poly-SUMOylation of the identified DDR factors. Collectively, these data point towards coordinated group-modification of DDR factors by SENP6.

**Fig. 1 | SENP6 is required for maintaining genomic stability. a** Cartoon demonstrating deconjugation of SUMO2/3 chains by SENP6 and the persistence of SUMO2/3 chains on a substrate in the absence of SENP6. S: SUMO2/3. **b-e** U2OS were treated with SENP6 (siSENP6) or nontargeting (NTP) siRNAs for 72 h and with or without 1 μM ML792 for 24 h, $n = 3$. **b** Representative images from γH2AX (magenta) and SUMO2/3 (green) immunofluorescence staining. **c** Quantification of γH2AX and SUMO2/3 foci. Data are shown as pooled cells (from left, $n = 314,314,260,329$ cells) (grey), superimposed by the median number of foci per experiment (orange, blue and purple). Significance was determined by two-sided one-way ANOVA with Tukey's multiple comparison test. **d** Quantification of γH2AX and SUMO2/3 colocalizing foci, expressed as a percentage of cells with 5 or more colocalizing foci. Significance was determined with a two-sided unpaired *t*-test. **e** Left panel: representative images of cells with micronuclei. Right panel: average percentage of cells with ≥1 micronuclei and with ≥1 γH2AX⁺ micronuclei. Significance was determined with a two-sided unpaired *t*-test. **c–e** Independent

experiments are visualized by unique symbols. **f** PrestoBLUE cell viability in U2OS treated with siSENP6 or NTP siRNAs and exposed to ionizing radiation (IR), camptothecin (CPT), mitomycin C (MMC) and hydroxyurea (HU), $n = 3$ with every condition set up in triplicate. Cell viability is expressed as percentage, with untreated conditions set at 100%. Significance was determined with a two-sided unpaired *t*-test at each dose. **g** Representative images of a single cell from γH2AX and SENP6 or GFP immunofluorescence staining of parental U2OS and U2OS stably expressing wildtype GFP-SENP6 (GFP-SENP6$_{WT}$) and catalytic dead GFP-SENP6 (GFP-SENP6$_{CD}$) 10 min after UVA laser micro-irradiation, n(U2OS) = 3; n(GFP-SENP6$_{WT}$) = 4; n(GFP-SENP6$_{CD}$) = 3. 140-172 cells were imaged per experiment for parental U2OS; 38-32 cells GFP-SENP6⁺ were imaged per experiment for GFP-SENP6$_{WT/CD}$. Percentage of GFP-SENP6⁺ cells with GFP-SENP6 at a UVA laser track was quantified. **c, d, e, f, g** Data in graphs are shown as mean ± SD. **b, e, g** Scale bar: 10 μm.

## SENP6 counteracts polySUMOylation in response to genotoxic stress

For multiple of the identified SENP6 substrates, increased SUMOylation upon different DNA damage treatments has previously been shown by mass-spectrometry or immunoblotting[14,32–35]. For example, HU treatment increased total SUMO conjugation levels and the SUMOylation levels of BRCA1, BARD1, BLM, and MDC1[35]. To investigate whether SENP6 does not only maintain these proteins in a hypo-SUMOylated state under unstressed conditions but also during active DNA damage signaling, we determined SUMOylation levels by immunoblotting after 2 mM HU treatment for 24 h in control shRNA-transduced and SENP6 shRNA-transduced U2OS cells expressing His10-SUMO2. As expected, SENP6 knockdown or HU treatment alone led to an increase in total SUMO conjugation levels, as well as the SUMOylation levels of BRCA1, BARD1, BLM, and MDC1 (Fig. 4a). Albeit subtle and variable, we observed further increases in the SUMOylation levels of these proteins in HU-treated SENP6 shRNA-transduced cells, compared to the single treatment conditions (Fig. 4a, b). Because of a substantial increase in SUMOylation levels of some of the proteins in the single treatment conditions, the margin for detecting further increases could be limited. Our data suggest that SENP6 also counteracts polySUMOylation of DDR proteins during DNA damage signaling.

## Depletion of SENP6 and RNF4 increases BRCA1, BARD1 and BLM SUMOylation

The paradigmatic signaling function for polySUMO2/3 chains is to target proteins for VCP-mediated extraction and degradation by the proteasome through STUbLs. RNF4 was previously demonstrated to operate in DDR signaling by ubiquitination of a variety of SUMOylated DDR proteins[36]. Furthermore, knockdown of SENP6 was shown to activate RNF4 through an increase in polySUMO2/3 substrates, ultimately causing its auto-ubiquitination and degradation in a negative feedback loop[37]. Among the validated proteins from our SENP6-target screen, we observed lower levels of BRCA1, BLM and CtIP in SENP6-depleted cells, which were partially restored by blocking the proteasome with MG132 (Fig. 2c, Supplementary Fig. 4b). Knockdown with shRNA SENP6#1 did not affect RNF4 levels, while shRNA SENP6#2 led to a minor decrease (Supplementary Fig. 4c). In agreement, down-regulation of some DDR proteins in SENP6-depleted cells using mass-spectrometry was described previously[24]. To further investigate the potential crosstalk between SENP6 and RNF4 in regulating DDR proteins, we co-depleted SENP6 and RNF4, and determined SUMOylation levels of a subset of SENP6 targets we identified. Co-depletion of SENP6 and RNF4 led to a further increase in total SUMO conjugation levels compared to SENP6 depletion alone, suggesting increased SUMOylation of at least a subset of proteins (Fig. 5a, b). Immunoblotting for BRCA1, BARD1, BLM showed further increases in their SUMOylation levels when both RNF4 and SENP6 were co-depleted; for

BRCA1 and BLM this was also visible in total lysates (Fig. 5a, b). We did not observe an increase in total MDC1 SUMOylation levels, but perhaps a minor shift towards higher molecular-weight conjugates. This could point towards coordinated regulation of these proteins by both SENP6 and RNF4, although we cannot formally exclude that knockdown of RNF4 indirectly affects SUMOylation levels, since SUMO E3 ligases were also identified as RNF4 targets[14].

## SENP6 depletion results in persistence of SUMO2/3 at DNA damage sites

We next wanted to investigate whether regulation of the SUMOylation levels of these proteins by SENP6 was also reflected in their kinetics at sites of DNA damage. To address this, we performed γH2AX and SUMO2/3 immunofluorescence after UVA laser micro-irradiation. We observed more SUMO2/3 at the DNA damage tracks 1 h-8 h after laser micro-irradiation in SENP6 knockdown cells compared to control cells (Fig. 6a, b, Supplementary Fig. 4d, e). In control-treated cells, SUMO2/3 was present in lower quantities at the DNA damage tracks at early timepoints (1 h and 2 h) and then gradually diminished, as has been described previously[30]. Similarly, more SUMO2/3 foci and SUMO2/3 and γH2AX colocalization were observed in SENP6-depleted cells after IR (Fig. 6c, d, Supplementary Fig. 4f). In control-treated cells, there was a timely induction in SUMO2/3 foci, which gradually diminished at later timepoints (4 h and 8 h). However, in SENP6-depleted cells, SUMO2/3 foci persisted over time as well as their colocalization with γH2AX. Immunofluorescence for γH2AX and BRCA1 revealed more BRCA1 foci and colocalization with γH2AX at early timepoints in SENP6-depleted cells (Fig. 6e, f). Similarly, there was a higher number of 53BP1 foci and colocalization with γH2AX at all timepoints in SENP6-depleted cells (Fig. 6g, h). Collectively, these data suggest that SENP6 is required for timely recruitment and clearance of SUMOylated DDR proteins at sites of DNA damage.

In agreement with others, we found that depleting SENP6 with siSENP6 for 72 h or 96 h led to co-depletion of RNF4, which could be rescued with the SUMO-E1 inhibitor ML792 (Supplementary Fig. 5a)[24,37]. To evaluate a potential combined effect from depleting both proteins, we compared co-depletion of SENP6 and RNF4 (siSENP6 cells) with depletion of RNF4 only (siRNF4 cells) for γH2AX and SUMO2/3 immunofluorescence. In contrast to siSENP6 cells, siRNF4 cells did not show a significant increase in γH2AX foci (Supplementary Fig. 5b, c). Depleting RNF4 did lead to a significant increase in SUMO2/3 foci, albeit to a much lesser extent (Supplementary Fig. 5c). We observed a similar effect in cells treated with IR. At all timepoints post-treatment, the number of SUMO2/3 foci was higher in siSENP6 cells compared to siRNF4 cells (Supplementary Fig. 5d). These data suggest that the observed effects related to genomic stability are not indirectly caused by the depletion of RNF4, but by SENP6 and RNF4 crosstalk, and potentially RNF4-independent regulation by SENP6.

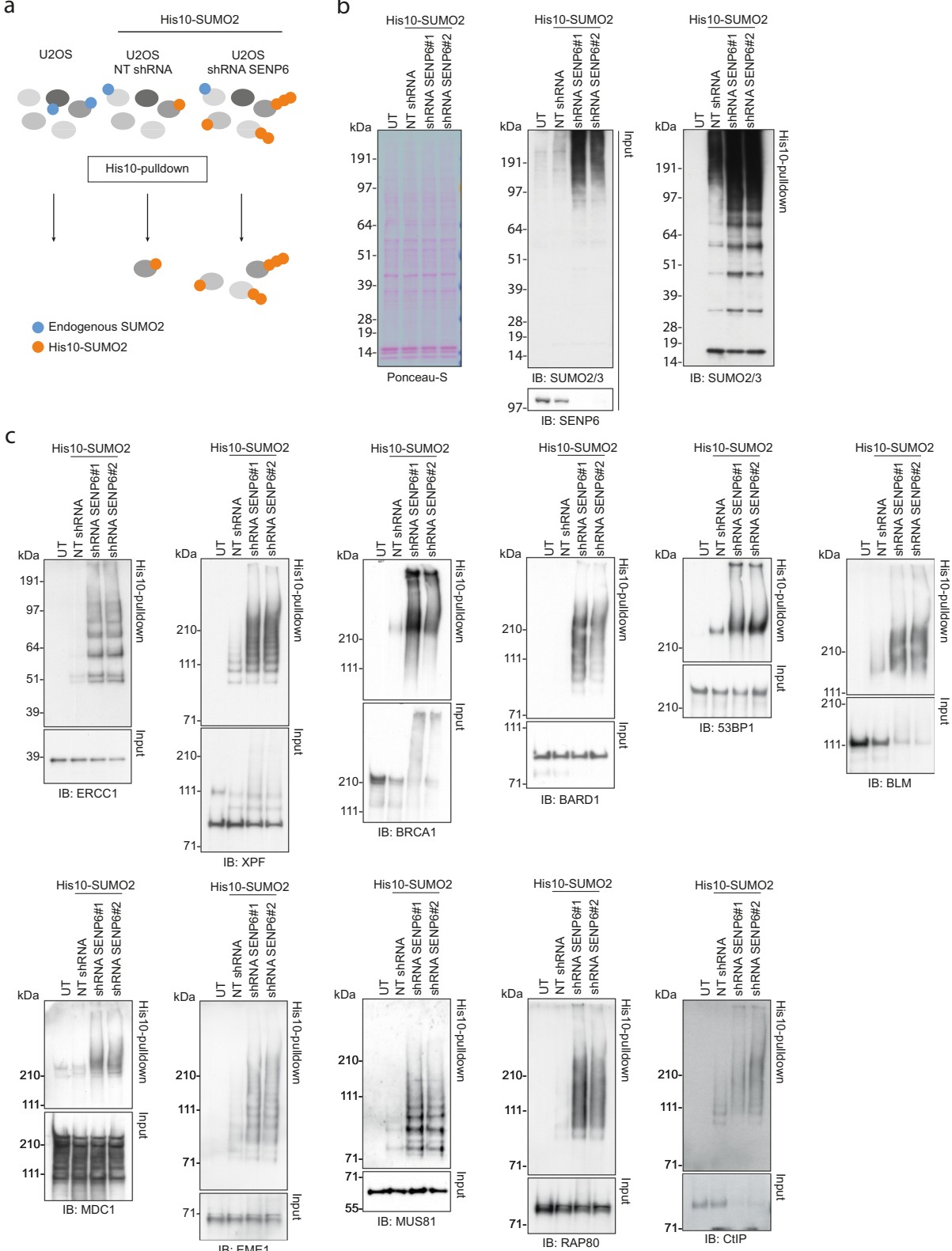

**Fig. 2 | SUMO2/3 polymers accumulate on DDR proteins in absence of SENP6.**
**a** Schematic representation of the His10-SUMO2 purification methodology combined with shRNA-mediated knockdown of SENP6. NT shRNA: nontargeting control shRNA. shRNA SENP6: SENP6-targeting shRNA. **b**, **c** U2OS stably expressing His10-SUMO2 were infected with lentiviruses encoding shRNAs against SENP6 (shRNA SENP6#1 and #2) or a nontargeting control shRNA (NT shRNA). Cells were lysed three days post-infection and His10-SUMOylated proteins were enriched by means of Ni-NTA pulldown. Total lysates (input) and His10-pulldown elutions were analyzed by immunoblotting for SENP6 and SUMO2/3, $n = 4$ (**b**) and a selection of DDR proteins identified in our previous mass-spectrometry screen, $n = 2$ (**c**). Equal loading of total lysates was verified by Ponceau-S staining. Representative blots are shown.

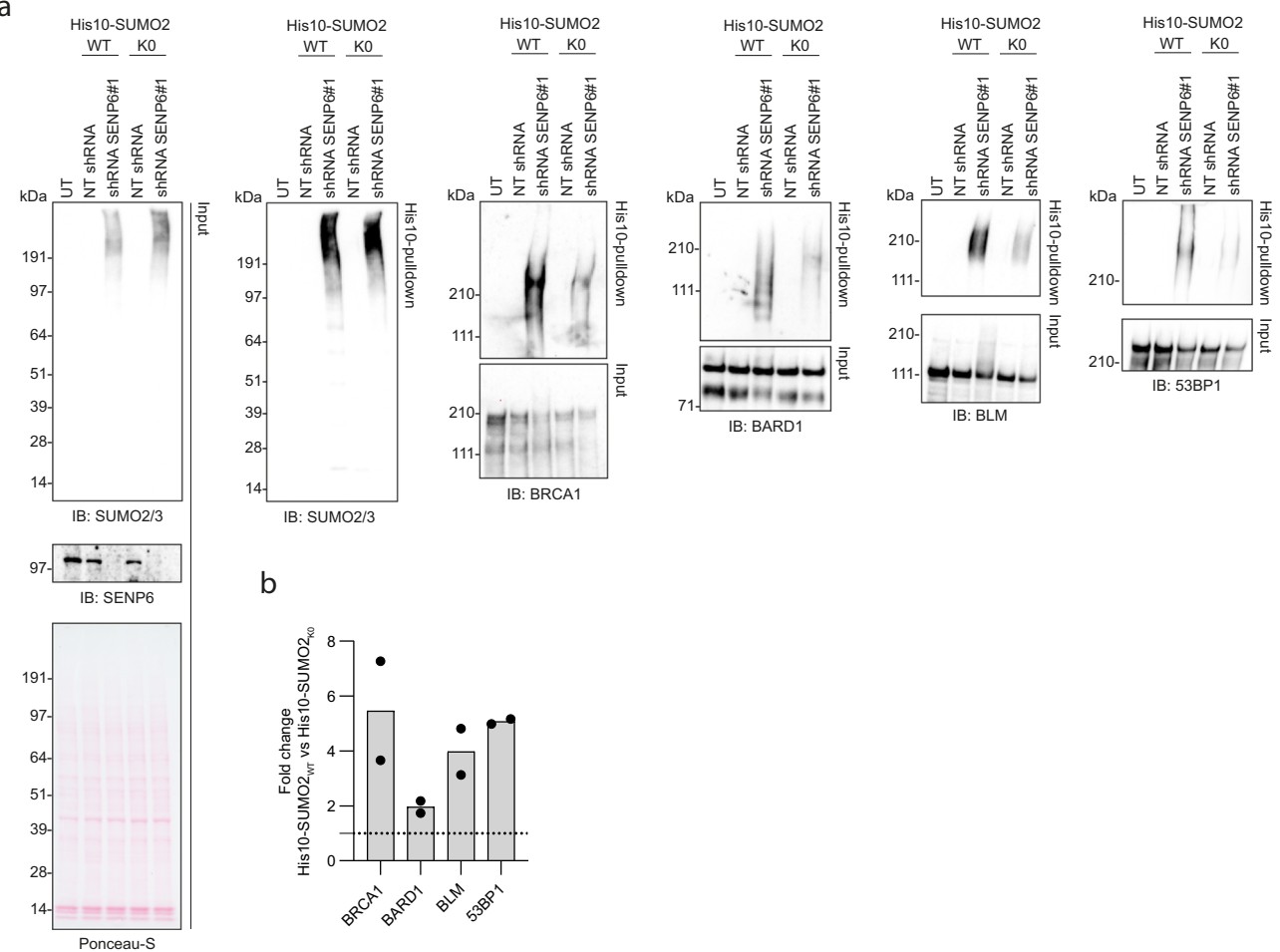

**Fig. 3 | SENP6 counteracts polySUMOylation of DDR proteins rather than multi-monoSUMOylation. a, b** U2OS stably expressing His10-SUMO2$_{WT}$ or His10-SUMO2$_{K0}$ were infected with lentiviruses encoding shRNAs against SENP6 (shRNA SENP6#1 and #2) or a nontargeting control shRNA (NT shRNA). Cells were lysed three days post infection and His10-SUMOylated proteins were enriched by means of Ni-NTA pulldown. Total lysates (input) and His10-pulldown elutions were analyzed by immunoblotting for SENP6, SUMO2/3 and a selection of DDR proteins, $n = 2$ (**a**). Equal loading of total lysates was verified by Ponceau-S staining. Representative blots are shown. Fold change in the SUMOylation levels of the DDR proteins was quantified by dividing the intensity in His10-SUMO2$_{WT}$ cells by the intensity in His10-SUMO2$_{K0}$ cells, corrected for the fold change in total SUMO2/3 levels between these cell lines (**b**). Quantifications for each protein were done on samples from the same experiment and were run on the same blot. Bars represent the mean.

## PolySUMOylated DDR proteins accumulate in PML bodies

Since SUMO2/3, BRCA1, and 53BP1 foci also accumulated at other sites in the nucleus than sites of DNA damage, we further investigated their potential subcellular localization. PML bodies are nuclear bodies implicated in many cellular processes, including the DDR, though their exact function remains unclear[38,39]. PML bodies consist of multi-merized PML and partner proteins such as SP100 and DAXX. Multi-merizing PML protein undergoes SUMOylation and can thereby recruit nuclear proteins to the bodies through SUMO-SIM interactions. Similarly, the SUMOylation status of nuclear proteins might also influence interaction with PML and partner proteins through altered SUMO-SIM interactions, where increases in SUMOylation would likely enhance this.

SENP6 was previously reported to regulate PML body formation[26]. In agreement with this finding, PML and its partner protein SP100 were identified in our mass-spectrometry screen as SENP6 targets[23] and SUMO2/3 accumulated on both proteins upon knockdown of SENP6 (Fig. 7a). In control cells, we observed mono-, di- and tri-SUMOylated PML. Upon SENP6 knockdown, we observed an increase in higher-molecular weight SUMO conjugates, indicating a build-up of SUMO2/3 polymers on these proteins. These modified forms indeed represent SUMOylated forms of the protein since they could be reduced by the

SUMO-E1 inhibitor ML792 (Fig. 7b). Thus, SENP6 affects the SUMOy-lation status of PML and SP100, which can potentially affect the recruitment of other nuclear proteins. Consistent with previous literature[26], we found a significant increase in the number of PML bodies upon knockdown of SENP6 by immunofluorescence (Fig. 7b, Supplementary Fig. 6a). Inhibiting SUMOylation reversed the number of PML bodies in SENP6 knockdown cells to numbers observed in untreated control cells (Fig. 7c, Supplementary Fig. 6a), demonstrating that these effects are mediated through SUMOylation.

SUMO2/3 was previously shown to localize to PML bodies[26]. Indeed, we also found a considerable amount of SUMO2/3 to coloca-lize with PML in control conditions. Upon knockdown of SENP6, there was an increase in both PML bodies and SUMO2/3 foci, concomitant with an increase in PML bodies colocalizing with SUMO2/3 foci, expressed as a percentage of the total number of PML bodies per nucleus (Fig. 7d, Supplementary Fig. 6b). Inhibiting SUMOylation reversed this increase.

Subsequently, we studied the potential colocalization of the DDR proteins with PML bodies in the absence of SENP6. Upon knockdown of SENP6, we observed a significant increase in the percentage of PML bodies that colocalized with ERCC1 and XPF (Fig. 7d, Supplementary Fig. 6b). These effects were completely reversed by SUMO-E1

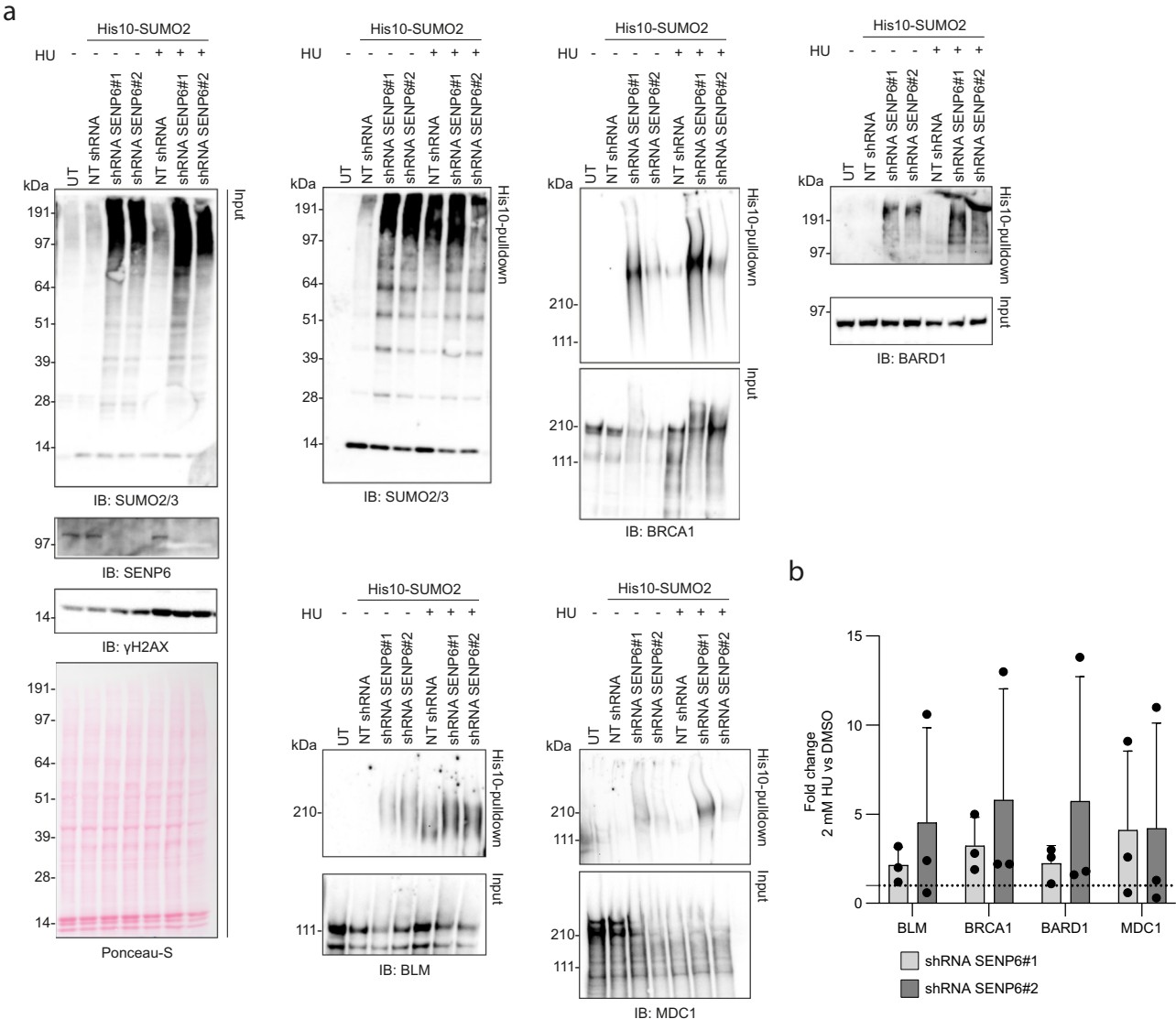

**Fig. 4 | SENP6 counteracts polySUMOylation of DDR proteins in response to hydroxyurea-induced genotoxic stress. a, b** U2OS stably expressing His10-SUMO2 were infected with lentiviruses encoding shRNAs against SENP6 (shRNA SENP6#1 and #2) or a nontargeting control shRNA (NT shRNA). Cells were treated with 2 mM hydroxyurea (HU) or DMSO for 24 h, before lysing cells three days post infection. His10-SUMOylated proteins were enriched by means of Ni-NTA pull-down. Total lysates (input) and His10-pulldown elutions were analyzed by immunoblotting for SENP6, SUMO2/3, and a selection of DDR proteins, $n = 3$ (**a**). Equal loading of total lysates was verified by Ponceau-S staining. Representative blots are shown. Fold change in the SUMOylation levels of the DDR proteins was quantified by dividing the intensity measured for the SENP6 knockdown cells treated with HU by the intensity for the DMSO-treated SENP6 knockdown cells (**b**). Quantifications for each protein were done on samples from the same experiment and were run on the same blot. Bars represent mean ± SD.

inhibition in SENP6 knockdown cells. BLM was described to localize to PML bodies during late S/G2 under control conditions[40]. We observed PML bodies to colocalize with BLM in the presence of SENP6, which further increased upon SENP6 knockdown (Fig. 7d, Supplementary Fig. 6b). The percentage of PML bodies that colocalized with BLM in the absence of SENP6 was partially reversed with SUMO-E1 inhibition. BRCA1 and 53BP1 also localized to PML bodies upon SENP6 knockdown, albeit to a lesser extent (Fig. 7d, Supplementary Fig. 6b). This was again reversed upon inhibition of SUMOylation. We confirmed that these DDR proteins were no longer SUMOylated upon SUMO-E1 inhibition with ML792 by immunoblotting, indicating that the observed effects are mediated by their SUMOylation (Supplementary Fig. 7a, b). RAP80, MUS81, and MDC1 showed minimal to no localization to PML bodies (Supplementary Fig. 7c). Taken together, these data show that some, but not all polySUMOylated DDR proteins accumulate in PML bodies in the absence of SENP6.

## SUMO2/3 polymers and SIM domains promote accumulation in PML bodies

Not all tested DDR proteins showed accumulation in PML bodies upon SENP6 knockdown. Additionally, other verified SENP6 substrates such as the CCAN protein CENP-C, did not localize to PML bodies (Supplementary Fig. 7c). This suggests that the sole presence of SUMO2/3 polymers on proteins is not sufficient for localization to PML bodies, but that other interactions are also required. Many DDR proteins have confirmed or putative SIM domains, which can potentially interact with SUMO moieties in PML bodies. Additionally, we have previously shown that mutating the functional SIM domain in SLX4 abrogates its localization to PML bodies and laser-track induced DNA damage sites[41].

We first investigated whether the DDR proteins were capable of SUMO binding by an in vitro binding assay with a SUMO2-trimer. Previously, we identified BLM, ERCC1, XPF, and RAP80 to bind to the SUMO2-trimer by mass-spectrometry screening[42]. Indeed BLM, XPF,

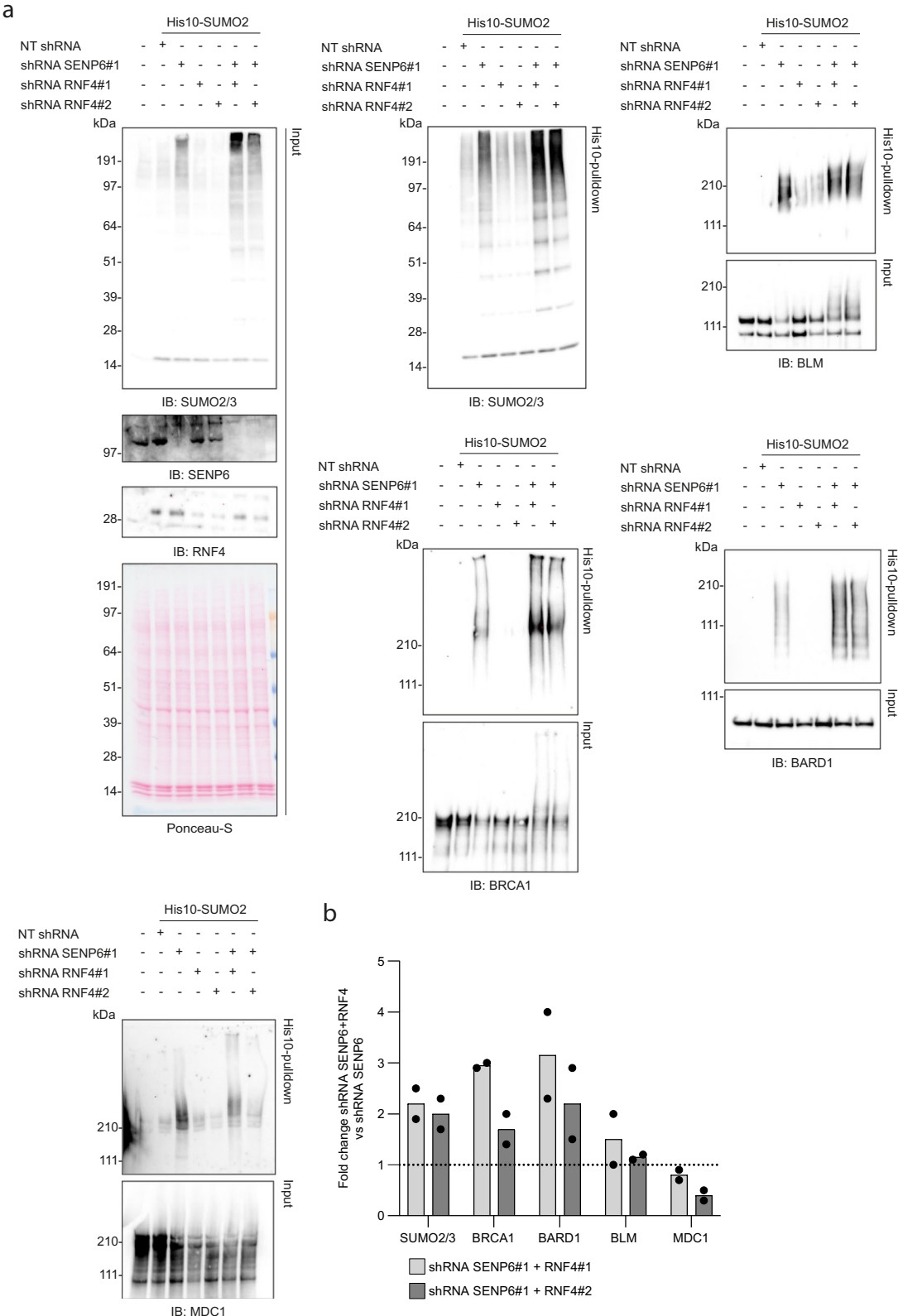

**Fig. 5 | Co-depletion of SENP6 and RNF4 further increases SUMOylation of BRCA1, BARD1 and BLM. a, b** U2OS stably expressing His10-SUMO2 were infected with lentiviruses encoding shRNAs against SENP6 (shRNA SENP6#1), RNF4 (shRNA RNF4#1 and shRNA RNF4#2) or a nontargeting control shRNA (NT shRNA). Cells were lysed three days post infection and His10-SUMOylated proteins were enriched by means of Ni-NTA pulldown. Total lysates (input) and His10-pulldown elutions were analyzed by immunoblotting for SENP6, RNF4, SUMO2/3 and a selection of DDR proteins, $n = 2$ (**a**). Equal loading of total lysates was verified by Ponceau-S staining. Representative blots are shown. Fold change in the SUMOylation levels of the DDR proteins was quantified by dividing the intensity measured for the SENP6 and RNF4 double knockdown cells by the intensity for the SENP6 knockdown cells (**b**). Quantifications for each protein were done on samples from the same experiment and were run on the same blot. Bars represent the mean.

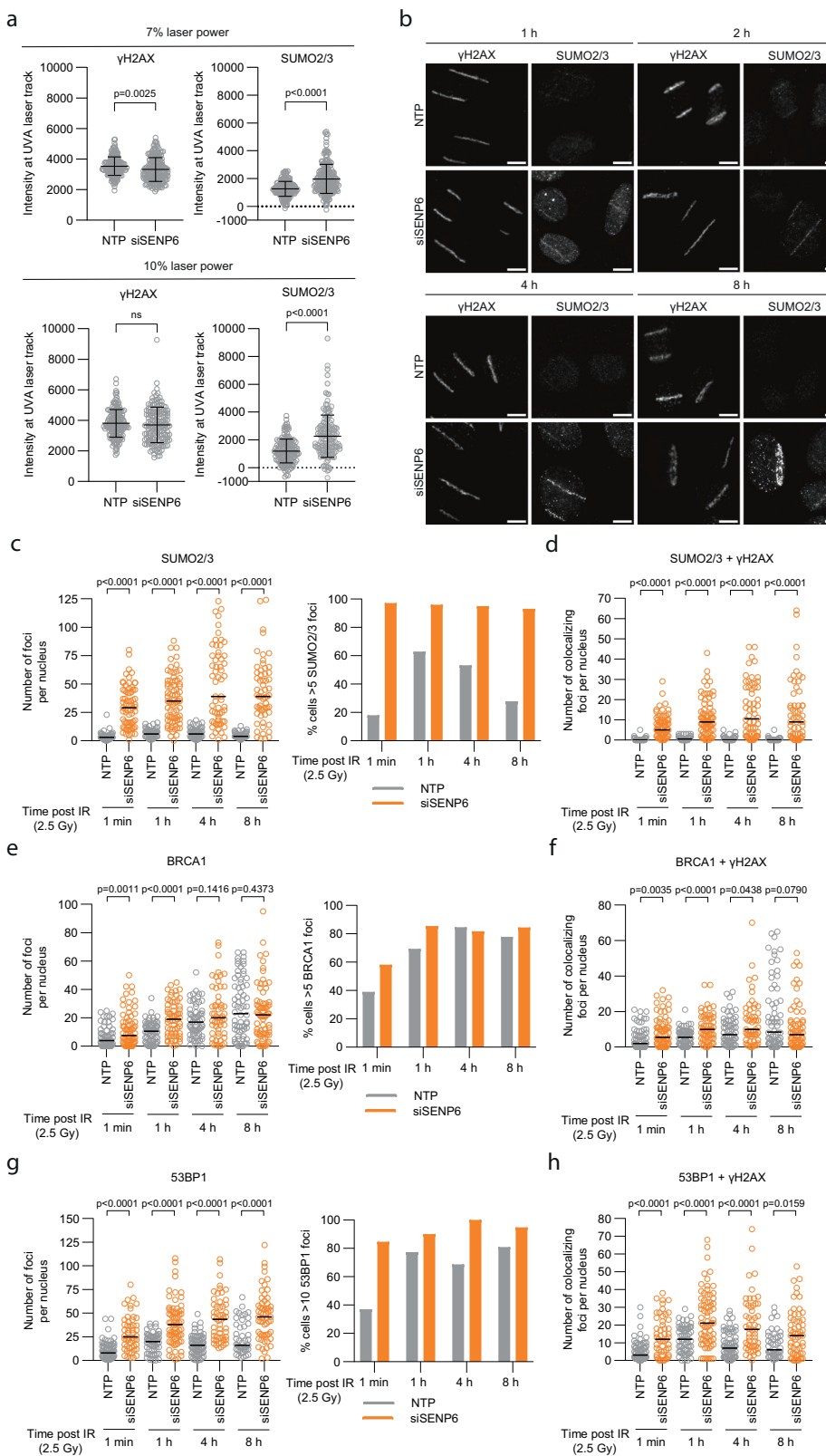

ERCC1, and RAP80 were capable of binding to the SUMO2-trimer in vitro, as well as BRCA1, 53BP1, and MUS81, although the ratio of unbound and bound protein varied (Fig. 8a). We could not correlate the extent in binding to the extent of PML body localization. However, this suggests that many of the DDR proteins have functional SIM domains and that SUMO-SIM interactions are likely functionally relevant for these proteins.

To further validate the importance of SIM domains on DDR proteins for localization to PML bodies, we generated a C-terminal GFP-tagged SIM mutant of ERCC1 for constitutive expression in U2OS. We identified two putative SIM domains in ERCC1, pSIM1 at position 23-26 (VIPL) and pSIM2 at position 101n = 103 (IIV), and generated both single and double mutants for these SIMs by mutating the long aliphatic residues to alanine (Fig. 8b, Supplementary Fig. 8a). Wildtype ERCC1

**Fig. 6 | SENP6 depletion leads to uncoordinated recruitment and persistence of SUMO2/3 at DNA damage sites. a** U2OS were treated with SENP6 siRNAs (siSENP6) or nontargeting siRNAs (NTP) for 72 h and exposed to UVA laser micro-irradiation. Cells were analyzed by γH2AX and SUMO2/3 immunofluorescence staining 1 h post irradiation. γH2AX and SUMO2/3 intensities at each UVA laser track were quantified. Every data point represents a single laser track (n(NTP 7%) = 189; n(siSENP6 7%) = 202; n(NTP 10%) = 167; n(siSENP6 10%) = 158). Error bars represent mean ± SD. Significance was determined with a two-sided unpaired *t*-test. **b** U2OS were treated as in (**a**), irradiated with a laser power of 7% and analyzed by γH2AX and SUMO2/3 immunofluorescence staining at the indicated timepoints. Representative images

of NTP and siSENP6 cells are shown. Scale bar: 10 μm. **c** U2OS were treated with siSENP6 or NTP siRNAs for 72 h before ionizing radiation (IR). Cells were analyzed at the indicated timepoints by γH2AX and SUMO2/3 immunofluorescence staining. The number of SUMO2/3 foci was quantified as well as the percentage of cells >5 SUMO2/3 foci (**c**) and the number of γH2AX and SUMO2/3 colocalizing foci per nucleus (**d**) (from left, n = 61,71,62,75,60,60,54,59 cells). **e**, **f** As cells in (**c**), but stained for γH2AX and BRCA1 (from left, n = 69,72,72,62,65,55,68,71 cells). **g**, **h** As cells in (**c**), but stained for γH2AX and 53BP1 (from left, n = 81,52,62,71,80,52,47,59 cells). **c–h** Lines in dot plots represent the median. Significance was determined with a two-sided Mann–Whitney U test at each timepoint.

and ERCC1$_{pSIM1}$ showed efficient binding to the SUMO2-trimer in vitro. However, ERCC1$_{pSIM2}$ and ERCC1$_{pSIM1+2}$ did not show binding, suggesting that pSIM2 is the functional SIM domain (from here on indicated as ERCC1$_{SIM}$) (Fig. 8c). To validate the importance of this SIM domain, we studied ERCC1$_{SIM}$ colocalization with PML bodies in the absence of SENP6. We found that ERCC1$_{SIM}$ completely lost the ability to accumulate in PML bodies compared to ERCC1$_{WT}$ after knockdown of SENP6 (Fig. 8d). In contrast to ERCC1$_{WT}$, ERCC1$_{SIM}$ hardly formed foci at all, also not at other sites in the nucleus, indicating that SUMO-SIM interactions could be of functional relevance to this protein (Supplementary Fig. 8b).

To further dissect the potential interplay between excessive SUMOylation and SUMO-SIM interactions, we generated U2OS stably expressing ERCC1$_{WT}$ and ERCC1$_{SIM}$ with a N-terminal GFP-3xSUMO2 fusion to mimic polySUMOylation (GFP-3xSUMO2-ERCC1$_{WT}$ and GFP-3xSUMO2-ERCC1$_{SIM}$) (Supplementary Fig. 8c) and studied colocalization with PML bodies. Adding 3xSUMO2 to ERCC1$_{WT}$ could not mimic the PML body localization that we observed for polySUMOylated ERCC1 after SENP6 knockdown (Fig. 8e). ERCC1$_{SIM}$ did not show any foci formation and PML localization, and adding 3xSUMO2 was not able to rescue this (Supplementary Fig. 8d).

Since ERCC1 functions as an endonuclease in complex with XPF and both proteins are confirmed SUMO substrates, we assessed whether this interaction was dependent on SUMO-SIM interactions. For this purpose, we purified ERCC1$_{WT}$-GFP and ERCC1$_{SIM}$-GFP from cells by trapping GFP and quantifying the amount of bound XPF. ERCC1$_{SIM}$ showed approximately a 50% reduction in XPF binding compared to ERCC1$_{WT}$ (Fig. 8f). Furthermore, inhibiting SUMOylation with SUMO-E1 inhibitor ML792 led to a similar reduction in XPF binding for ERCC1$_{WT}$, indicating that SUMOylation promotes this interaction. ML792 treatment did not affect total XPF levels and led to a slight increase in ERCC1-GFP$_{WT/SIM}$ levels (Supplementary Fig. 8e), indicating that the reduction in bound XPF could not be attributed to reduced total protein levels. Taken together, these data show that accumulation of ERCC1 and thereby consequent localization to PML bodies requires the presence of both SUMO2/3 polymers and SIM domains. Furthermore, we demonstrate that ERCC1-XPF complex formation is partly SUMO-dependent.

## PolySUMOylated DDR proteins can accumulate independently of PML

As of yet, there is no clear consensus on whether PML bodies themselves are actively involved in DNA repair and are recruited to DNA damage sites. Here, we observed that PML did not accumulate at γH2AX foci induced by knockdown of SENP6 (Fig. 9a, Supplementary Fig. 9a). However, in some cells, PML and γH2AX occasionally appeared to be juxtaposed, which could potentially be explained by the observation that telomerase-negative cells contain PML bodies that associate specifically with chromatin at telomeres and are involved in alternative lengthening of telomeres (ALT; ALT-PML bodies). Overall, this indicates that DDR proteins that reside in PML bodies after the knockdown of SENP6 cannot simultaneously function in DNA repair processes at double-strand breaks and form distinct subcellular condensates

To determine whether the aberrant accumulation of DDR proteins in PML bodies occurs independently of their premature accumulation and persistence at DNA damage sites, we generated U2OS cells deficient for PML (U2OS ΔPML) using a CRISPR-Cas9 approach (Supplementary Fig. 9b, c). U2OS ΔPML were viable, albeit proliferating at a slower rate (Supplementary Fig. 9d). We first evaluated the formation of γH2AX and DDR protein foci after SENP6 knockdown in U2OS ΔPML. The induction of DNA double-strand breaks was similar in U2OS ΔPML and parental U2OS after knockdown of SENP6 (Fig. 9b). Moreover, the formation of SUMO2/3 foci was still induced, albeit to a lesser extent than in parental U2OS, especially for U2OS ΔPML c3 (Fig. 9c). This reduction could perhaps be explained by the loss of SUMOylated PML and partner proteins. There was no change in SUMO2/3 localizing to DNA damage sites in the absence of PML bodies, as demonstrated by similar Pearson correlation coefficients for signal overlap (Fig. 9c). XPF foci formation induced by the knockdown of SENP6 was also comparable in U2OS ΔPML and parental U2OS, indicating that PML bodies are not causally involved in the formation of these aberrant condensates (Fig. 9d). There was no increase in XPF localizing to DNA damage sites in the absence of PML bodies (Fig. 9d). Immunofluorescence for XPF and SP100, another PML body component, in U2OS ΔPML showed that knockdown of SENP6 induced foci formation of both proteins (Supplementary Fig. 10a) as well as their colocalization (Supplementary Fig. 10b). On average 52% of the XPF foci colocalized with SP100. As in parental U2OS, treatment with the SUMO-E1 inhibitor ML792 prevented foci formation (Supplementary Fig. 10a–c). Besides XPF, knockdown of SENP6 also induced foci formation of BLM and 53BP1 in U2OS ΔPML in a SUMO-conjugation-dependent manner (Supplementary Fig. 10d). In line with these observations, we did not observe a rescue of the phenotype induced by knockdown of SENP6 in U2OS ΔPML. Consistent with the comparable induction of DNA double-strand breaks described above, cell viability of U2OS ΔPML was not improved compared to parental U2OS and was reduced even further (Fig. 9e). Taken together, these data suggest that poly-SUMOylation of the DDR proteins after knockdown of SENP6 causes these proteins to have a high propensity for forming protein condensates and that this occurs both at sites of DNA damage, leading to genomic instability, as well as at (PML) nuclear bodies. At sites of DNA damage this was reflected in premature accumulation and persistence of SUMO2/3, BRCA1, and 53BP1. Condensation in (PML) nuclear bodies was shown to be facilitated by multivalent SUMO-SIM interactions. Thus, SENP6 is required for the coordinated regulation of SUMOylated DDR proteins, governing their timely localization at DNA damage sites as well as their nuclear condensation state (Fig. 10).

## Discussion

Here, we addressed the role of the SUMO protease SENP6, responsible for deconjugation of SUMO2/3 polymers, in the DDR. Using a proteomics approach, we previously identified 180 proteins as potential SENP6 targets[23]. Of these, 51 unique proteins annotated to gene ontology processes involving cellular responses to DNA damage, of which 30 specifically to double-strand break repair. Immunoblotting experiments on a selection of these proteins confirmed build-up of SUMO2/3 polymers after SENP6 knockdown. Under steady-state

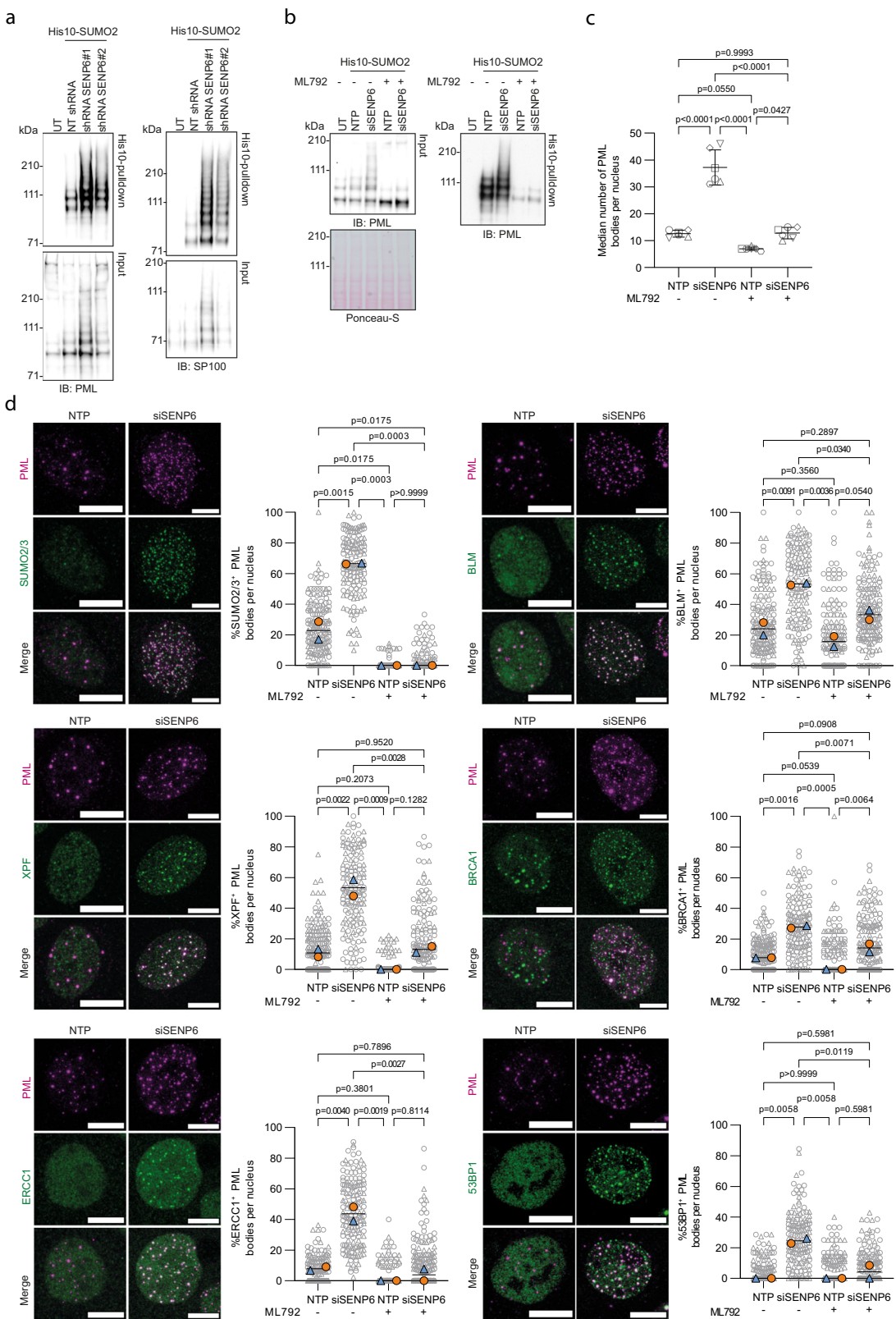

conditions in the presence of SENP6, SUMO2/3 polymers on DDR proteins are virtually absent, suggesting that their (de-)conjugation is a highly dynamic process. Further illustrating their dynamic nature is our observation that we could reverse the build-up of SUMO2/3 polymers on DDR proteins induced through SENP6 knockdown by blocking SUMO-conjugation with ML792. We show that SENP6 does not only maintain these proteins in a hypoSUMOylated state under unstressed conditions but also counteracts their polySUMOylation in response to genotoxic stress. We evaluated only a single genotoxic agent at a single timepoint, but the overlap of our identified SENP6 targets with previous SUMO mass-spectrometry screens using different DNA damage-inducing treatments and cell cycle stages suggests that SENP6 likely also counteracts the polySUMOylation of additional DDR proteins during DNA damage signaling to the ones validated here.

**Fig. 7 | PolySUMOylated DDR proteins accumulate in PML bodies. a** Samples from Fig. 2 were also analyzed by immunoblotting for PML and SP100, $n = 2$. Same loading was used and verified by Ponceau-S staining. Representative blots are shown. **b** U2OS stably expressing His10-SUMO2 were treated with SENP6 siRNAs (siSENP6) or nontargeting siRNAs (NTP) for 96 h and with or without 1 μM ML792 for 24 h. Cells were lysed and His10-SUMOylated proteins were enriched by means of Ni-NTA pulldown. Total lysates (input) and His10-pulldown elutions were analyzed by immunoblotting for PML, $n = 2$. Equal loading of total lysates was verified by Ponceau-S staining. A representative blot is shown. **c** U2OS were treated as in (**b**) and analyzed by immunofluorescence staining for PML, $n = 6$ visualized by unique symbols. Data are shown as median number of PML bodies per nucleus, with quantification ranging from 44-130 cells per condition. Error bars represent mean ± SD. Significance was determined by two-sided one-way ANOVA with Tukey's

multiple comparison test. **d** Cells from (**c**) were also analyzed by immunofluorescence staining for PML (magenta) and the indicated DDR proteins (green), $n = 2$. Representative images of a single cell are shown. Scale bar: 10 μm. Colocalization is quantified as percentage of PML bodies that colocalize with a DDR protein per nucleus. Data are shown as pooled cells (SUMO2/3 from left, $n = 170, 154, 192, 220$; XPF from left, $n = 186, 157, 179, 158$ cells; ERCC1 from left, $n = 142, 169, 163, 209$; BLM from left, $n = 175, 153, 152, 178$; BRCA1 from left, $n = 208, 153, 144, 166$; 53BP1 from left, $n = 168, 150, 195, 169$ cells) (grey), superimposed by the median percentage per experiment (blue and orange). Independent experiments are visualized by unique symbols. Lines represent the mean. Significance was determined by two-sided one-way ANOVA with Tukey's multiple comparison test.

Previous work described a functional role for SENP6 in the DDR by counteracting SUMOylation of the DDR proteins EXO1, RPA70, and the Fanconi Anemia proteins FANCI and FANCD2[16,17,25]. In another recent proteomics screen, SENP6 was identified as a key regulator of proteins involved in the ATR-Chk1 DNA damage checkpoint and sister-chromatid cohesion[24]. Collectively, we and others provide a comprehensive set of data supporting regulation of the DDR by SENP6 through protein-group modification. This is in line with the model proposed by Psakhye and Jentsch[10,11], where SUMOylation targets groups of functionally related proteins and acts synergistically to regulate nuclear processes including the DDR.

SENP2 and SENP7 were shown to regulate DNA repair by timely deconjugation of SUMOylated proteins at damaged chromatin[21,22]. We occasionally observed wildtype and catalytic dead SENP6 at UVA laser-induced DNA damage tracks, but not the endogenous protein. Others previously described localization of exogenously expressed catalytically inactive HA-SENP6 at FANCI foci after MMC treatment but failed to observe localization at laser-irradiated sites[16]. Moreover, cellular fractionation assays previously found SENP6 in both the chromatin and nucleoplasmic fractions under control conditions, and aphidicolin treatment did not lead to a re-localization of the protein detectable by immunoblotting[24]. Since SENP6 is also critical in regulating other cellular processes and is not exclusively or predominantly a DDR protein, it is possible that SENP6 localization remains diffuse despite the presence of DNA damage. Moreover, enrichment of SENP6 at sites of DNA damage could be very local and transient, making it difficult to reliably and consistently detect this re-localization by cellular fractionation assays or standard immunofluorescence, especially for the endogenous protein. In support of local and transient enrichment of SENP6 at the chromatin, enrichment of SENP6 was detected with iPOND mass-spectrometry at stalled-replication forks 10 min after induction, but not yet at 5 and no longer at 15 min[43]. The exact timing and localization of deSUMOylation by SENP6 of DDR proteins in response to DNA damage requires further investigation.

SUMO localizes to sites of DNA damage[29,30] and SUMO2/3 was previously shown to colocalize with γH2AX foci induced by SENP6 knockdown[17]. Here, we show that there is an excess of SUMO2/3 at DNA damage sites induced by IR or UVA laser micro-irradiation after depletion of SENP6. This occurs at early timepoints after DNA damage induction and persists over time. Additionally, DDR proteins BRCA1 and 53BP1 were also present in excess at IR-induced DNA damage sites at early timepoints, and like SUMO2/3, 53BP1 persisted at these sites over time. This suggests that SENP6 is required for the timely localization and clearance of SUMOylated DDR proteins at DNA damage sites.

In the DDR, a recurrent concept is the timely regulation of protein turnover by SUMO-mediated clearance of DDR proteins from the chromatin (reviewed in ref. 7). The paradigmatic signaling function for SUMO2/3 polymers is to stimulate the STUbL pathway, leading to ubiquitination and proteasomal degradation of proteins. Indeed, SENP6 was shown to counterbalance SUMO2/3-polymer-induced

dimerization and activation of RNF4, promoting both substrate ubiquitination and RNF4 autoubiquitination, followed by their degradation[37]. We observed lower levels of BRCA1, BLM, and CtIP in SENP6-depleted cells and partially rescued these levels by blocking of the proteasome. In agreement, downregulation of some DDR proteins in SENP6-depleted cells using mass-spectrometry was described previously[24]. Furthermore, co-depletion of SENP6 and RNF4 further increased the SUMOylation levels of BRCA1, BARD1 and BLM compared to SENP6 or RNF4 depletion alone, further pointing towards coordinated regulation of DDR proteins by both SENP6 and RNF4. An increase in BLM SUMOylation upon co-depletion of RNF4 and SENP6 was also demonstrated by others, under unstressed conditions and in the presence of hydroxyurea[34].

In agreement with others, we observe co-depletion of RNF4 with siRNA-mediated knockdown of SENP6, potentially limiting proteolytic degradation of polySUMOylated proteins[24,37]. Comparing co-depletion of both proteins in siSENP6 cells with single depletion of RNF4, we found that the spontaneous increase in γH2AX and SUMO2/3 foci, as well as the localization and persistence of SUMO2/3 at IR-induced DNA damage sites, are not indirectly caused by depletion of RNF4, but by SENP6 and RNF4 crosstalk, and potentially RNF4-independent regulation by SENP6. PARPi sensitivity of SENP6-deficient lymphoma cells was shown to be predominantly, but not completely RNF4-dependent, suggesting that RNF4-independent regulation of some DDR proteins by SENP6 is also important for maintaining genomic integrity[24]. Which SENP6 substrates are antagonistically regulated by SENP6 and RNF4 and to what extent potential RNF4-independent modes of regulation by SENP6 play a role in the DDR is still an outstanding question. Because of the tight relationship between SENP6 and RNF4 levels, dissecting potential RNF4-indepenent regulation of DDR proteins by SENP6 is challenging.

There is growing evidence supporting a critical role for SUMOylation in the dynamics and compositional control of phase-separated molecular condensates (reviewed in ref. 44). PML nuclear bodies are a quintessential example of nuclear condensates that appear to form through phase separation and recent in vitro studies provide striking evidence that this is driven by SUMO polymers and SUMO-SIM interactions[45]. By controlling SUMOylation levels of PML through SUMO (de-)conjugation, PML bodies can recruit SIM-containing proteins and regulate their composition. Indeed, SENP6 was previously identified as a regulator of PML bodies[26]. PML was also identified as SENP6 substrate in our proteomics screen[23] and we confirmed build-up of SUMO2/3 polymers after SENP6 knockdown on PML as well as SP100, an important partner protein, and PML body component. We hypothesize that the build-up of SUMO2/3 polymers enhanced SUMO-SIM interactions between PML and DDR proteins by providing excess binding moieties. Supporting our hypothesis, we found increased colocalization of SUMO2/3, ERCC1, XPF, BLM, BRCA1, and 53BP1 foci with PML bodies after SENP6 knockdown. We found that this was induced through excessive SUMOylation of the DDR proteins, by preventing SUMO2/3 polymer build-up and reversing this effect with

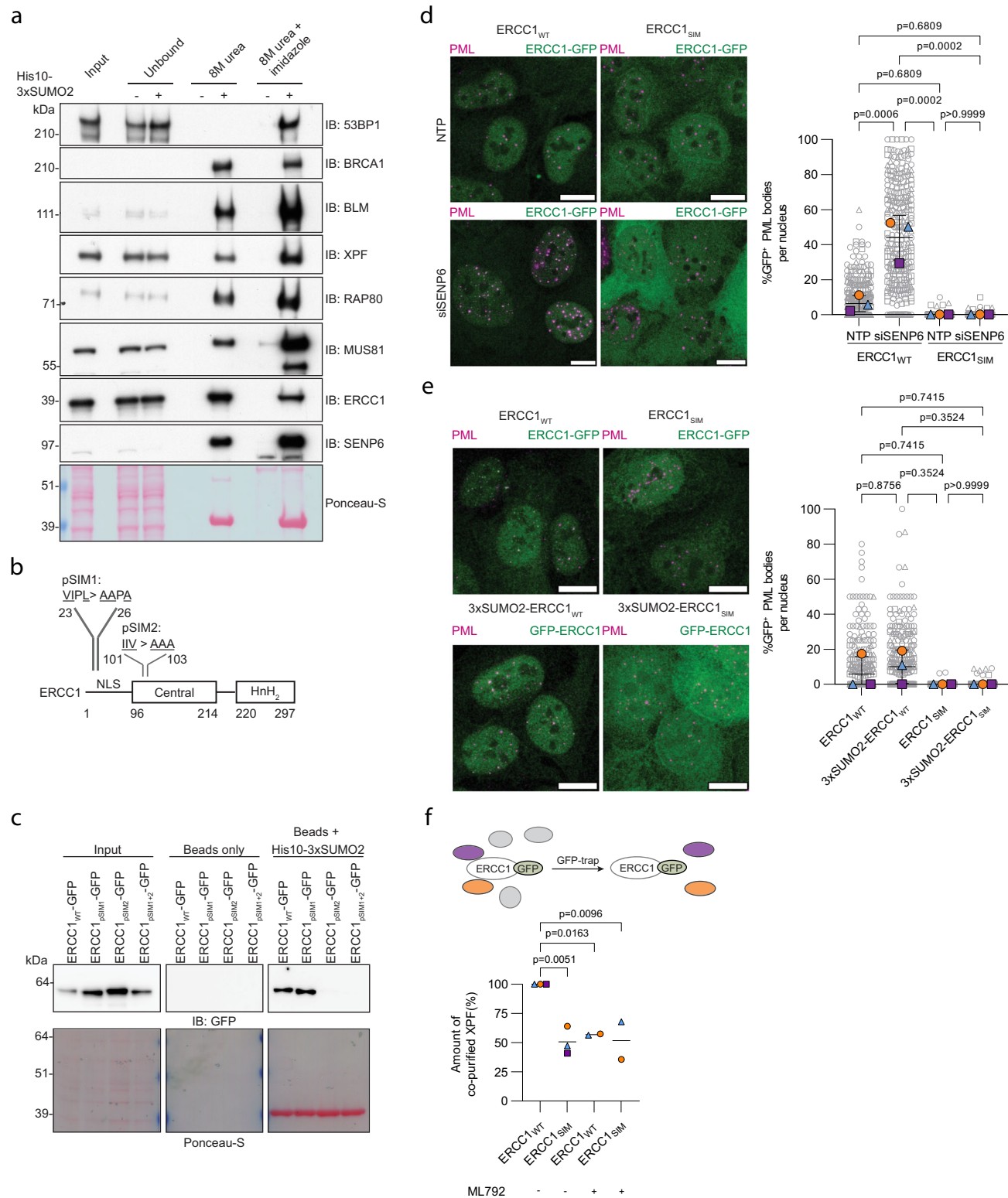

the SUMO-E1 inhibitor ML792. ERCC1, XPF, BLM, BRCA1, and 53BP1 bound non-covalently to a SUMO2-trimer, indicating their ability to participate in SUMO-SIM interactions with SUMO2/3 polymers. Further supporting that these molecular condensates are governed by SUMO-SIM interactions, we showed that SIM-defective ERCC1, incapable of binding to SUMO2/3, can no longer form foci that localize to PML bodies when SUMO2/3 polymer build-up is induced by knockdown of SENP6. Thus, non-covalent SUMO binding via SIMs in DDR factors facilitates their accumulation in PML bodies. This contrasts with the

lack of SUMO binding of CCAN proteins that are also extensively SUMOylated in the absence of SENP6[23], but are not accumulating in PML bodies (this study). Furthermore, we found that these condensates can form independently of PML and potentially contain other nuclear body-associated proteins, like SP100. Following the model proposed by Banani et al.[45], this supports that the formation of these condensates is driven by multivalent SUMO-SIM interactions between nuclear proteins that have excess SUMO moieties and SIM domains available. Collectively, our data suggest that polySUMOylation of the

**Fig. 8 | SUMO2/3 polymers and SIM domains are required for accumulation of DDR proteins in PML bodies. a** U2OS cell lysates were analyzed for in vitro SUMO binding, $n = 2$. Total lysates (input), unbound fractions and bead elutions were analyzed by immunoblotting for the indicated DDR proteins. Representative blots are shown. **b** Schematic representation of ERCC1 and potential SIM domains. **c** Cell lysates from cell lines in Figure S8A were analyzed for in vitro SUMO binding, $n = 3$. Total lysates (input) and bead elutions were analyzed by immunoblotting for GFP. A representative blot is shown. **d** U2OS expressing ERCC1-GFP wildtype (ERCC1$_{WT}$) and ERCC1-GFP SIM mutant (ERCC1$_{SIM}$) were treated with SENP6 siRNAs (siSENP6) or nontargeting siRNAs (NTP) for 96 h and analyzed by immunofluorescence staining for PML (magenta) and GFP (green), $n = 3$. **e** U2OS expressing ERCC1-GFP wildtype (ERCC1$_{WT}$), ERCC1-GFP SIM mutant (ERCC1$_{SIM}$), GFP-3xSUMO2-ERCC1 wildtype (3xSUMO2-ERCC1$_{WT}$) and GFP-3xSUMO2-ERCC1 SIM mutant (3xSUMO2-ERCC1$_{SIM}$) were analyzed by immunofluorescence staining for PML (magenta) and GFP (green). **d**, **e** One representative image per condition is shown. Scale bar:

10 μm. The graph shows colocalization quantified as percentage of PML bodies that colocalizes with GFP per nucleus. Data are shown as cells pooled cells (from left in (**d**), $n = 301,297,296,261$ cells; from left in (**e**), $n = 248,256,251,255$ cells) (grey), superimposed by the median percentage per experiment (blue, orange, and purple). Independent experiments are visualized by unique symbols. Error bars represent mean ± SD. Significance was determined by two-sided one-way ANOVA with Tukey's multiple comparison test. **f** U2OS stably expressing ERCC1$_{WT/SIM}$ were treated with or without 1 μM ML792 for 24 h. ERCC1$_{WT/SIM}$ were purified and samples were and analyzed by immunoblotting for GFP and XPF. XPF levels were quantified and corrected for GFP-enrichment levels. Per the independent experiment, quantifications were done on samples that were run on the same blot. The amount of co-purified XPF is expressed as a percentage with the amount in untreated ERCC1$_{WT}$ set at 100%, $n = 2$ for ML792-treated, $n = 3$ for untreated. Lines represent the mean. Significance was determined by two-sided one-way ANOVA with Dunnett's multiple comparison test.

---

DDR proteins after knockdown of SENP6 causes these proteins to have a high propensity for forming protein condensates and that this results in uncoordinated recruitment and persistence of SUMOylated proteins at DNA damage sites as well as at (PML) nuclear bodies (Fig. 10). We show that the (PML) nuclear condensates are distinct subcellular condensates that do not localize to DNA damage sites. Recently, there is increasing evidence that DNA repair might also involve the formation of transient biomolecular condensates at sites of DNA damage (reviewed in ref. 46). Intriguingly, SENP6 and RNF4 were recently shown to regulate the assembly and disassembly of SLX4 condensates on chromatin that compartmentalizes DDR proteins in a SUMO-SIM-dependent manner, thereby facilitating DNA repair[47]. More research is necessary to further dissect the mechanism of nuclear condensation of DDR proteins through SUMO2/3 polymers and their physiological roles. It remains to be explored whether the persistence of SENP6 substrates at sites of DNA damage and their localization in distinct subcellular nuclear condensates are linked and occur in a parallel or perhaps sequential manner. Future research should focus on gaining more mechanistic insight into the modes of regulation by SUMO2/3 polymers and SENP6 for the functioning of distinct DDR proteins, further revealing the sequence of events in response to DNA damage at sites of damage, as well as upstream and downstream signaling.

SENP6 knockdown leads to a range of cell-cycle- and genome stability-related defects[17,22–24,26]. In accordance with previous work[17,24], we observed an increase in DNA double-strands breaks upon SENP6 knockdown, both in nuclei and micronuclei, suggesting micronuclei formation partly resulted from chromosome fragments generated through these double-strand breaks. The induction of DNA double-strand breaks through SENP6 knockdown could be reversed by treatment with the SUMO-E1 inhibitor ML792, illustrating that the observed DNA damage is induced through excessive SUMOylation. Additionally, we demonstrated increased sensitivity of cells after SENP6 knockdown to multiple DNA-damage-inducing treatments (HU, IR, CPT, and MMC), indicating widespread regulation of the DDR and maintenance of genome stability by SENP6. Regulation of genome stability by SENP6 was recently linked to tumor development and progression in MYC-driven B-cell lymphoma[27], highlighting that a better understanding of SUMO-polymer signaling and deconjugation may provide novel insights that could enable exploiting this pathway for anticancer therapy.

## Methods

A detailed list of antibodies, oligonucleotides, and recombinant DNA used in this study can be found in Supplementary Table 1.

### Cell culture

U2OS (ATCC® HTB-96™) (gender: female), HEK 293 T and HEK 293GP cells were cultured in Dulbecco's modified Eagle's medium (DMEM) (high glucose, pyruvate, Gibco™) supplemented with 10% FCS and

100 U/ml penicillin and 100 μg/ml streptomycin (Gibco™). U2OS cells stably expressing His10-SUMO$_{WT}$ and His10-SUMO$_{K0}$ were established previously[48]. U2OS cells stably expressing inducible wildtype or catalytic dead GFP-SENP6 fusion constructs were established previously[23]. Expression was induced with 1 μg/ml doxycycline treatment for 24 h. For inhibition of the proteasome, cells were treated for 6-8 h with 10 μM MG132 (Sigma-Aldrich). For inhibition of the SUMO E1 enzyme, cells were treated with 1 μM ML792 (407886, MedKoo Biosciences Inc.) for 24 h. For induction of genotoxic stress, cells were treated for 24 h with 2 mM hydroxyurea (Sigma-Aldrich). Cells were exposed to the indicated doses of IR and UVA laser micro-irradiation (described below), and processed after the indicated timepoints. Cells were checked routinely for mycoplasma and found to be negative.

### Generating PML knockout cell lines

U2OS cells were co-transfected with an expression vector containing Cas9-2A-GFP (pSpCas9(BB)-2A-GFP (PX458); Addgene #48138) and pU6-gRNA/PGK-Puro-2A-BFP (Human Sanger Arrayed Whole Genome Lentiviral CRISPR library from Sigma-Aldrich), containing the following gRNA sequence: 5'-ATCCAAGAAAGCCAGCCCAGAGG-3'. Transfected cells were selected on 1 μg/ml puromycin for three days. Subsequently, cells were plated at low density, cultured, and individual clones were isolated. PML knockout was validated by microscopy and immunoblotting. The absence of Cas9 integration/stable expression was confirmed by GFP microscopy and immunoblotting for Cas9.

### Generating ERCC1 SIM mutant-cell lines

pDONR223 encoding the open reading frame (ORF) of ERCC1 was obtained from the MISSION® TRC3 Human ORF Collection (Sigma-Aldrich). Putative SIMs were mutated by replacing isoleucine, leucine or valine residues with alanine using two-step PCR-mediated mutagenesis. The following primers were used to create the SIM mutants. pSIM1: 5'-GCCAGCAAGGAAGAAATTTGCGGCACCCGCCGACGAGGAT GAGGTCCCTC-3' (forward) and 5'-GAGGGACCTCATCCTCGTCGGC GGGTGCCGCAAATTTCTTCCTTGCTGGC-3' (reverse). pSIM2: 5'-CCG GGGCAAAATCCAACAGCGCCGCTGCGAGCCCTCGGCAGAGGGGCA-3' (forward) and 5'-TGCCCCTCTGCCGAGGGCTCGCAGCGGCGCTGTTG GATTTTGCCCCGG-3' (reverse). Mutants were verified by sequencing the ORF. Subsequently, ERCC1$_{WT}$, ERCC1$_{pSIM1}$, ERCC1$_{pSIM2}$ and ERCC1$_{pSIM1+2}$ were cloned into pBABE-puro-C-term-GFP using Gateway® cloning, and retrovirus was made to generate cells stably expressing these protein-constructs. SIM mutants were validated by in vitro SUMO binding. To create ERCC1 with a tri-SUMO2 (3xSUMO2) fusion, ERCC1$_{WT}$ and ERCC1$_{SIM}$ were amplified by PCR with primers containing EcoRI and NotI restriction enzyme sites. These PCR products were digested with EcoRI and NotI, and ligated into pDONR207-3X-ΔN11-3X-SUMO2-ΔGG-CENPW, linearized by EcoRI and NotI restriction enzyme digest, thereby replacing CENPW with ERCC1. 3xSUMO2-ERCC1$_{WT/SIM}$ was subsequently cloned into pBABE-puro-N-term-GFP using Gateway®

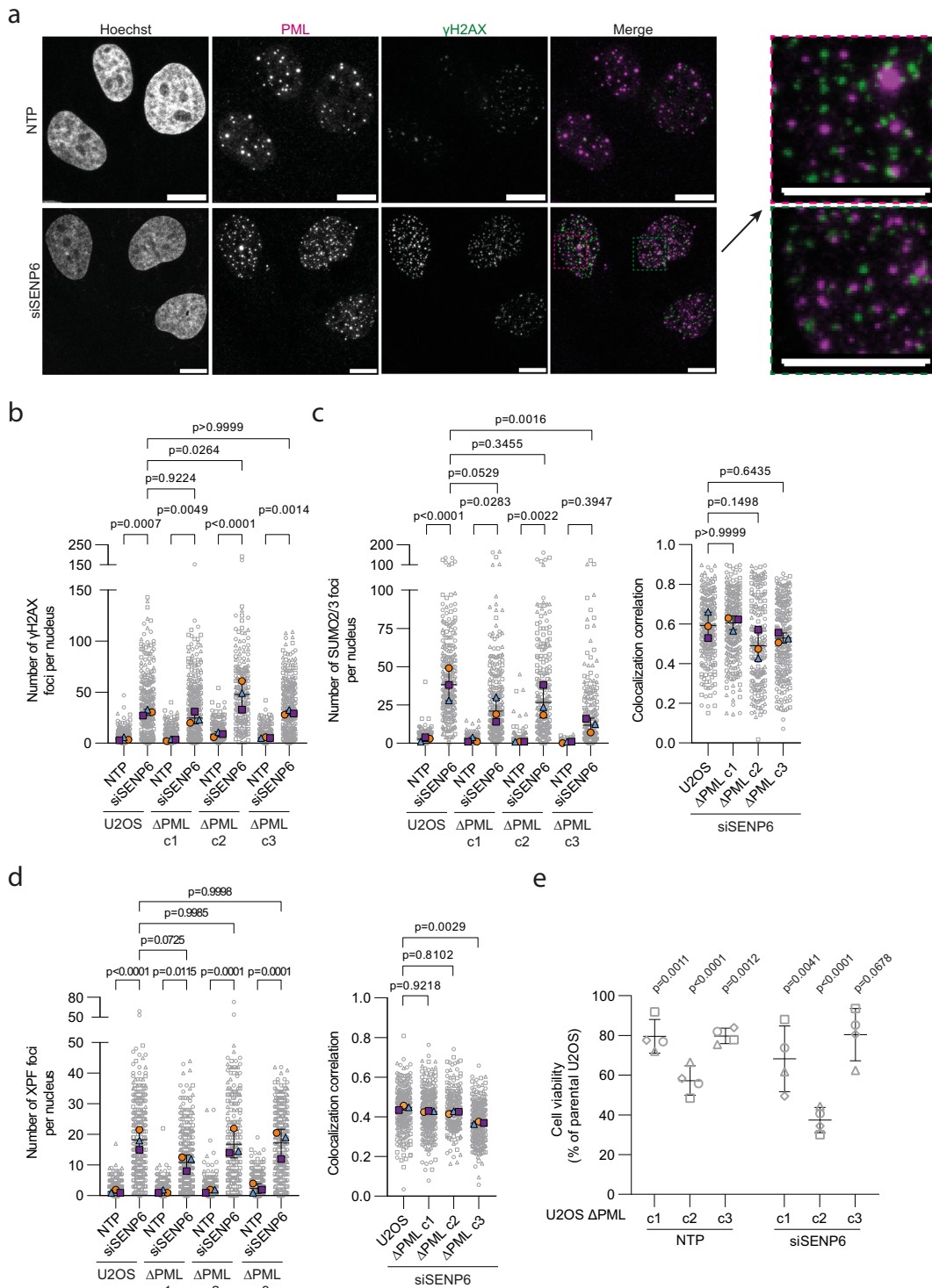

cloning and retrovirus was made to generate cells stably expressing these protein-constructs.

### Lentivirus production and transduction

Lentivirus was produced in HEK 293 T cells transfected with third-generation lentiviral packaging plasmids (pCMV-VSVG, pMDLg-RRE and pRSV-REV) and plasmids encoding SENP6 shRNA (Sigma-Aldrich Mission® shRNA library; TRC-004103 and TRC-004104), RNF4 shRNA (Sigma-Aldrich Mission® shRNA library; TRC-272668 and TRC- 284821) or nontargeting control shRNA (Sigma-Aldrich Mission® shRNA library; SHC002). For shRNA-mediated knockdown, cells were transduced with lentivirus at a MOI of 3 in DMEM containing 8 µg/ml polybrene. The transduction medium was replaced after 24 h and cells were lysed 72 h after infection.

### Retrovirus production and transduction

For stable expression of ERCC1$_{WT}$-GFP, ERCC1$_{pSIM1}$-GFP, ERCC1$_{pSIM2}$-GFP, ERCC1$_{pSIM1+2}$-GFP, GFP-3xSUMO2-ERCC1$_{WT}$ and GFP-3xSUMO2-ERCC1$_{SIM}$, HEK 293GP cells were transfected with pBABE-puro-[insert]-GFP or pBABE-puro-GFP-[insert] together with a plasmid encoding the viral envelope VSV-G protein. Cells were transduced with the harvested retrovirus in DMEM containing 8 µg/ml polybrene. Transduced cells

**Fig. 9 | PolySUMOylated DDR proteins can accumulate in nuclear condensates independently of PML. a** U2OS were treated with a pool of SENP6 siRNA (siSENP6) or nontargeting siRNAs (NTP) for 96 h and analyzed by immunofluorescence staining for γH2AX (green) and PML (magenta). Amount of cells imaged ranged from 62-88 per condition. Representative images are shown, *n* = 3 for α-PML mouse (Ms) and α-γH2AX rabbit (Rb), and *n* = 2 for α-PML Rb and α-γH2AX Ms containing. Scale bar: 10 μm. **b–d** U2OS were treated with siSENP6 or NTP siRNAs for 96 h and analyzed by immunofluorescence staining for γH2AX and SUMO2/3, and γH2AX and XPF, *n* = 3. Data are shown as number of foci or colocalization correlation in individual nuclei pooled (from left in **b**, **d**, *n* = 316,281, 310,301,236,202,286,273 cells; from left in **c**, *n* = 354,242,316,259,258,205,306,256 cells) (grey), superimposed by the median number of foci or correlation per experiment (blue, orange and purple). Independent experiments are visualized by

unique symbols. Error bars represent mean ± SD. Significance was determined by two-sided one-way ANOVA with Tukey's multiple comparison test. **b**, **c**, **d** Left panels: Comparisons between NTP- and siSENP6-treated cells for each cell line, and siSENP6-treated U2OS ΔPML cells versus parental U2OS are shown. **c**, **d** Right panels: Comparisons between siSENP6-treated U2OS ΔPML cells versus parental U2OS are shown. **e** PrestoBLUE cell viability assay of U2OS and U2OS ΔPML cells treated with siSENP6 or NTP siRNAs. Viability of U2OS ΔPML cells five days after transfection is shown as a percentage, with parental U2OS in NTP and siSENP6-treated conditions set to 100% viability. Data are shown as mean ± SD, *n* = 4 visualized by unique symbols. Significance between U2OS ΔPML cells and parental U2OS was determined by two-sided one-way ANOVA with Dunnett's multiple comparison test for NTP and siSENP6-treated conditions separately.

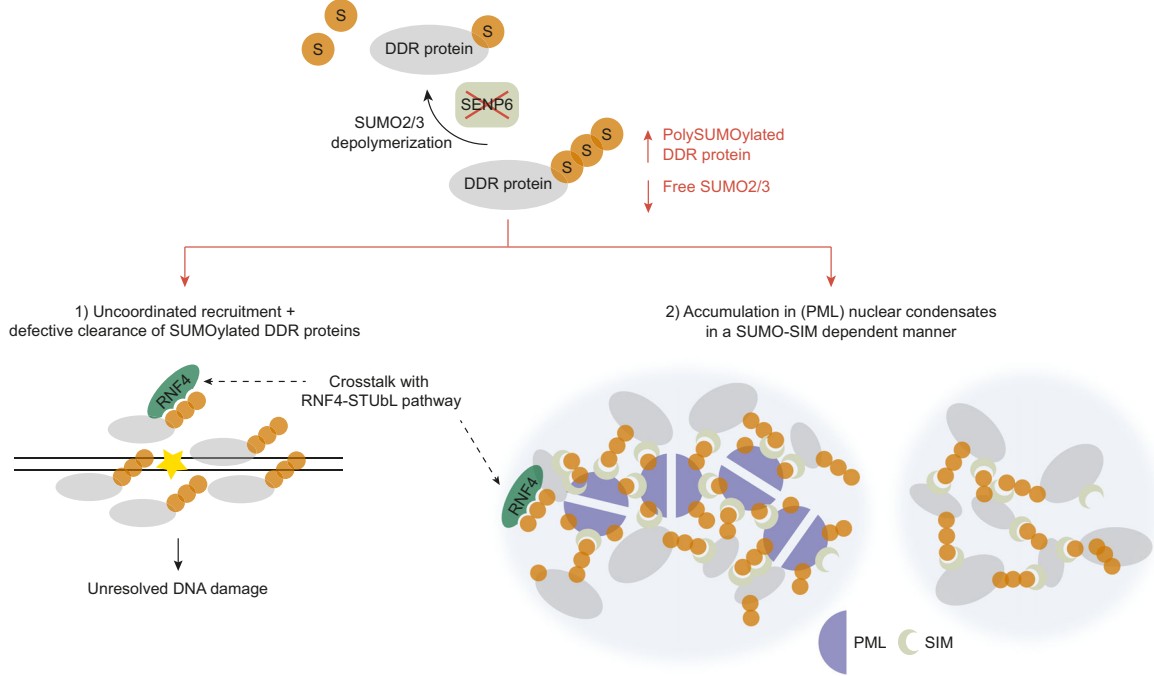

**Fig. 10 | SENP6 regulates localization and nuclear condensation of DNA damage response proteins by group deSUMOylation.** Model indicating how SENP6 regulates localization and nuclear condensation of DDR proteins by group deSUMOylation. Depletion of SENP6 leads to uncoordinated localization and persistence of SUMOylated DDR proteins at DNA damage sites, as well as their accumulation in (PML) nuclear bodies. S: SUMO2/3.

were selected in a medium containing 1 μg/ml puromycin. Expression of ERCC1$_{WT}$-GFP, ERCC1$_{pSIM1}$-GFP, ERCC1$_{pSIM2}$-GFP, ERCC1$_{pSIM1+2}$-GFP, GFP-3xSUMO2-ERCC1$_{WT}$ and GFP-3xSUMO2-ERCC1$_{SIM}$ was confirmed by immunoblotting for GFP and ERCC1, and GFP microscopy.

### siRNA-mediated knockdown

For siRNA-mediated knockdown, U2OS cells were transfected with 10 nM SMARTpool ON-TARGETplus SENP6 siRNA (Dharmacon; L-006044-00), ON-TARGETplus RNF4 siRNA (Dharmacon; J-006557-08) or SMARTpool ON-TARGETplus nontargeting siRNA (Dharmacon; D-001810-10) mixed with Opti-MEM™ (Gibco) and Lipofectamine RNAi-MAX (Thermo Fisher Scientific, 13778075). The transfection medium was replaced after 24 h and cells were processed after transfection as indicated.

### His10-SUMO2 purification

His10-SUMO2 conjugated proteins were purified as described previously[31]. In brief, U2OS cells stably expressing His10-SUMO2 were harvested in ice-cold PBS. For total protein lysates, a small fraction of cells were lysed in 2% SDS, 1% NP-40, 50 mM Tris pH 7.5, and 150 mM NaCl. The remainder was lysed in 25 pellet volumes of 6 M

guanidine-HCl, 100 mM Na$_2$HPO$_4$/NaH$_2$PO$_4$, 10 mM Tris, buffered at pH 8.0. Lysates were snap-frozen in liquid nitrogen and stored at −80 °C or directly processed. Cells were thawed at room temperature and sonicated twice for 10 s at 30 W. Protein concentration was determined by bicinchoninic acid (BCA) Protein Assay Reagent (Thermo Fisher Scientific) and lysates were equalized. Subsequently, lysates were supplemented with 5 mM β-mercaptoethanol and 50 mM imidazole pH 8.0. Pre-washed Ni-NTA beads (Qiagen, 30210) were added to the lysates and incubated overnight at 4 °C. Ni-NTA beads were washed with wash buffer 1–4, respectively; Wash buffer 1: 6 M Guanidine-HCl, 100 mM sodium phosphate, 10 mM Tris pH 8.0, 10 mM imidazole pH 8.0, 5 mM β-mercaptoethanol, 0.2% Triton X-100. Wash buffer 2: 8 M urea, 100 mM sodium phosphate, 10 mM Tris pH 8.0, 10 mM imidazole pH 8.0, 5 mM β-mercaptoethanol, 0.2% Triton X-100. Wash buffer 3: 8 M urea, 100 mM sodium phosphate, 10 mM Tris pH 6.3, 10 mM imidazole pH 7.0, 5 mM β-mercaptoethanol, 0.2% Triton X-100. Wash buffer 4: 8 M urea, 100 mM sodium phosphate, 10 mM Tris pH 6.3, 5 mM β-mercaptoethanol, 0.1% Triton X-100. Purified proteins were eluted twice in one bead volume of 7 M urea, 100 mM sodium phosphate, 10 mM Tris pH 7.0, and 500 mM imidazole pH 7.0.

## Immunostaining for microscopy

Cells were grown on coverslips and fixed in 4% paraformaldehyde for 15 min and washed with PBS. Subsequently, cells were permeabilized with 1% Triton X-100 in PBS for 15 min at room temperature and washed twice with PBS and once with PBS containing 0.05% Tween-20. Where indicated, cells were pre-extracted with 0.2% Triton X-100 in PBS for 3 min on ice, before fixation. Cells were then blocked in 0.1 M Tris-HCl pH 7.5, 0.15 M NaCl, 5 mg/ml Boehringer Blocking Reagent (TNB) for at least 10 min at room temperature. Primary antibodies were diluted in TNB and incubated for 1 h at room temperature. Cells were washed five times with PBS containing 0.05% Tween-20 and incubated with a secondary antibody in TNB for 1 h at room temperature. Cells were washed five times with PBS containing 0.05% Tween-20. DNA was stained with Hoechst 33342 (Thermo Fisher Scientific; 5–10 μg/ml) for 20 min. Cells were then dehydrated in a stepwise manner with 70% ethanol for 1 min, 90% ethanol for 1 min and 100% ethanol for 1 min. Coverslips were mounted on microscopy slides with anti-fade prolong gold (Thermo Fisher Scientific).

## Microscopy imaging and analysis

Imaging was performed on a Leica SP8 confocal microscope with LAS X software. For the imaging in Figs. 1, 6, Supplementary Figs. 1, 2, 5, 9c, and 10, snapshots were taken around the equatorial plane. For the imaging in Figs. 7–9, Supplementary Figs. 6–8, and 9a, -15 z-stacks were acquired at 0.3 μM steps. All images were acquired using a 63x objective, 1.4 NA. Multiple image fields were acquired from different sections of the coverslips, typically consisting of -15-20 cells per field, in order to reach sufficient numbers for quantification and to capture heterogeneity. Imaging was performed using a fixed laser power for each antibody staining across technical and biological replicates. Images were analyzed using ImageJ (v1.53f51). We used the BIC Macro Toolkit created by the Universität Konstanz Bioimaging Centre for foci quantification and colocalization. For z-stack images, maximal intensity projections were generated. In brief, areas of nuclei were selected based on Hoechst staining. Foci were then identified in each channel based on the Find Maxima function using a fixed noise value for each antibody across replicates. Two foci were considered as colocalizing when the distance between them was at most 2 pixels. In Figs. 1d, 6c, e, g, and Supplementary Fig. 1c, colocalization was quantified by calculating the percentage of nuclei that had 5 or more colocalizing foci. In Figs. 6d, f, h, and Supplementary Fig. 10b colocalization was quantified by number of colocalizing foci. Elsewhere, colocalization was quantified by dividing the number of colocalizing foci by the total number of foci of one of the foci channels as a reference per nucleus, expressed as a percentage. Signal overlap of the two foci channels estimated by the Pearson correlation coefficient was sometimes also plotted to illustrate colocalization as determined by a different quantification method.

## UVA laser micro-irradiation microscopy

Cells were grown on 18 mm coverslips and sensitized with 10 μM 5′-bromo-2-deoxyuridine (BrdU) for 24 h before UVA laser micro-irradiation. Cells were washed with PBS and culture medium was replaced by CO2-independent Leibovitz's L15 medium (Thermo Fisher) supplemented with 10% FBS. For micro-irradiation, cells were placed in a live-cell imaging chamber set to 37 °C on a Zeiss Axio Observer microscope using a 63x (1.4 NA) oil-immersion objective. Laser micro-irradiation was performed using a diode-pumped solid-state 355 nm laser integrated into a UGA-42 Caliburn system (Rapp OptoElectronic). Laser power was set to 7% or 10% (Figs. 1g and 6b: 7%; Fig. 6a: 7 and 10%). Following micro-irradiation, cells were incubated for the indicated timepoints at 37 °C and subsequently pre-extracted with ice-cold 0.25% Triton X-100 in CSK-B buffer (10 mM HEPES, 300 mM Sucrose, 100 mM NaCl, 3 mM MgCl2 pH 7.4) for 2 min on ice, followed by fixation with 4% paraformaldehyde for 15 min at room temperature.

Subsequently, cells were stained and mounted as described above. Imaging for γH2AX and SUMO2/3 immunostaining was performed on a Zeiss Axio Imager D2. Imaging for SENP6 localization was performed on a Leica SP8 confocal microscope. For all imaging, snapshots were taking around the equatorial plane. ImageJ (v1.53f51) was used for γH2AX and SUMO2/3 signal quantification at the UVA laser tracks. For this, average pixel intensity was measured in the irradiated area, in the nucleoplasm outside of the irradiated area and in a region in the same field not containing cells. The signal intensity at a laser track was subtracted with the signal intensity in the nucleoplasm, corrected for the signal intensity in the background of the image, to quantify the relative amount of protein at the laser tracks.

## Cell viability assays

U2OS or U2OS ΔPML cells were transfected for 24 h with 10 nM SMARTpool ON-TARGETplus SENP6 siRNA (Dharmacon; L-006044-00) or SMARTpool ON-TARGETplus nontargeting siRNA (Dharmacon; D-001810-10) mixed with Opti-MEM™ (Gibco) and Lipofectamine RNAiMAX (Thermo Fisher Scientific, 13778075). Cells were then harvested and re-seeded at low density in triplicate in 96-well plates. For DNA damage-inducing experiments, cells were treated with ionizing radiation (IR), hydroxyurea (HU), camptothecin (CPT), or mitomycin C (MMC) at the indicated doses and concentrations. CPT and MMC were removed after 1 h, HU was removed after 24 h. Five days post-treatment, cell viability was measured using PrestoBLUE cell viability reagent (Thermo Fisher Scientific) according to the manufacturer's protocol. For U2OS ΔPML cells, viability was measured five days post re-seeding. Absorption was measured at 544/590 nm after incubating for 1 h at 37 °C.

## Electrophoresis and immunoblotting

Cells were lysed in 2% SDS, 1% NP-40, 50 mM Tris pH 7.5, and 150 mM NaCl, unless stated otherwise. Proteins were separated on 4-12% gradient gels (Bold, Thermo Fisher Scientific) using MOPS buffer or 3-8% gradient gels (NuPage, Thermo Fisher Scientific) using Tris-Acetate buffer. Proteins were subsequently transferred onto Amersham Protran Premium 0.45 NC Nitrocellulose blotting membrane (GE Healthcare), using a submarine system. Membranes were stained with Ponceau-S to visualize total protein and blocked with PBS containing 0.05% Tween-20 and 8% milk powder. Subsequently, membranes were stained with primary and secondary antibodies diluted in PBS containing 0.05% Tween-20 and 8% milk powder. ImageJ (v1.53f51) was used for signal quantification of protein bands, where applicable.

## In vitro SUMO binding assay for endogenous proteins

His10-tagged SUMO2 trimer was produced as described before[49]. Two aliquots of 100 μl Ni-NTA bead slurry were washed three times with 50 mM Tris pH 7.5, 150 mM NaCl, 0.5% NP-40, 50 mM imidazole pH 7.0. To one aliquot, 100 μg of recombinant His10-SUMO2 trimer was added; the other aliquot of beads was used as a negative control. Subsequently, beads were incubated for 2 h at 4 °C while rotating and then washed again three times with 50 mM Tris pH 7.5, 150 mM NaCl, 0.5% NP-40, 50 mM imidazole pH 7.0. Cell pellets from ten 15 cm dishes of U2OS cells were lysed in 1 ml lysis buffer (50 mM Tris pH 7.5, 150 mM NaCl, 0.5% NP-40, 50 mM imidazole pH 7.0). Lysates were sonicated 2 times for 10 s, split over 1.5 ml microcentrifuge tubes and centrifuged for 1 h at 4 °C at 16,000 x g. Input samples were taken from the supernatant and the remaining supernatant was added to the Ni-NTA beads with the His10-SUMO2 trimer or beads only and incubated 2 h at 4 °C while rotating. After incubation, an unbound control sample was taken and beads were washed three times for 10 min with lysis buffer (50 mM Tris pH 7.5, 150 mM NaCl, 0.5% NP-40, 50 mM imidazole pH 7.0), followed by three times for 10 min with wash buffer (50 mM Tris pH 7.5, 150 mM NaCl), including a tube change after each second wash. Proteins were eluted from the beads

with 100 μL 8 M urea, 50 mM Tris pH 7.5 for 30 min at 4 °C at 1200 r.p.m. The second elution was performed with 100 μl 8 M urea, 50 mM Tris pH7.5, 500 mM imidazole pH 7.0 for 30 min at 4 °C at 1200 r.p.m.

### In vitro SUMO binding assay for SIM validation
The in vitro SUMO binding assay as described above was modified for SIM validation. Cell pellets from three 15 cm dishes of U2OS cells stably expressing ERCC1$_{WT}$-GFP, ERCC1$_{pSIM1}$-GFP, ERCC1$_{pSIM2}$-GFP or ERCC1$_{pSIM1+2}$-GFP were lysed in 1 ml lysis buffer. Lysates were equalized after sonication and clarification by BCA. Aliquots of 30 μl of Ni-NTA bead slurry were used and 30 μg of recombinant His10-SUMO2 trimer. Proteins were eluted from the beads with 30 μl of the respective buffers.

### Colony forming assay
Cells were seeded in duplicate at low density in 6-well plates and grown for 14 days. Then, cells were washed once with PBS and fixed in ice-cold 100% methanol for 20 min at −20 °C. Cells were stained with 0.5% crystal violet for 30 min at room temperature. Plates were rinsed, dried and images were taken. Crystal violet was resolubilized with 100% methanol and quantified in triplicate in a 96-well plate by measuring absorbance at 595 nm in a plate reader.

### ERCC1-GFP protein purification from cells
U2OS cells expressing ERCC1-GFP fusion constructs as described above were lysed in lysis buffer (20 mM Tris pH 7.5, 150 mM NaCl, 0.5% Triton X-100, 1 mM MgCl$_2$, 20 mM NEM and cOmplete™ EDTA-free Protease Inhibitor Cocktail). 100 Units of Benzonase (Merck Millipore) were added and samples were vortexed and incubated for 1 h at 4 °C in a rotating wheel. Lysates were then centrifuged for 1 h at °4 C at 16,000 x $g$ and equalized by BCA. Input samples were taken from the supernatant. The remaining supernatant was added to GFP-Trap Agarose (Chromotek) in lo-bind Eppendorf tubes, which were pre-washed twice with lysis buffer. Samples were incubated for 1.5-2 h at 4 °C in a rotating wheel. The unbound fraction was taken before washing the GFP-trap Agarose three times with wash buffer (20 mM Tris pH 7.5, 150 mM NaCl, 0.5% Triton X-100, 1 mM MgCl$_2$, 20 mM NEM and cOmplete™ EDTA-free Protease Inhibitor Cocktail). Proteins were eluted of the beads by adding 2x LDS sample buffer and boiling for 5 min at 95 °C. Beads were sedimented by centrifugation and supernatant was analyzed by immunoblotting.

### Gene ontology and STRING network analysis
STRING network analysis and Gene Ontology (GO) enrichment analysis of enriched proteins after SENP6 knockdown (Source proteomics data: ProteomeXchange Consortium via the PRIDE partner repository PXD011963)[23] was performed in Cytoscape 3.8.0 with a STRING confidence score of 0.7 or higher. GO enrichment analysis was filtered for GO processes. The top enriched GO processes related to the DNA damage response were selected and STRING networks were generated for the genes annotated to each of these GO processes.

### Reporting summary
Further information on research design is available in the Nature Portfolio Reporting Summary linked to this article.

## Data availability
The data that support this study are available from the corresponding author upon request. Source data are provided with this paper.

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

## Acknowledgements

We thank Edwin Willemstein for preparation of the samples in Fig. 6a. We thank Wouter Wiegant and Haico van Attikum for instructions on use of the microscope and analysis for the UVA laser micro-irradiation experiments. This study was supported by the Dutch Research Council (NWO) (724.016.003 to A.C.O.V.).

## Author contributions

A.C.O.V. conceived and supervised the project. A.C.O.V. and L.A.C. designed experiments. L.A.C. conducted experiments, analyzed data, and made the figures. I.J.d.G. and M.V.d.V. conducted experiments. L.A.C. and A.C.O.V. wrote the manuscript.

## Competing interests

The authors declare no competing interests.
