## [Peer Review File · Nature Communications]

SENP6 regulates localization and nuclear condensation of DNA damage response proteins by group deSUMOylationReviewers' Comments:

Reviewer #1:

Remarks to the Author:

Claessens & Vertegaal build on previous work that identify the SUMO protease SENP6 as an important regulator of various groups of proteins including factors of the DDR network. They show that spontaneous DSB repair signalling is increased in SENP6 depleted cells, resulting a surprisingly mild sensitivity to DSB inducing agents. They confirm several important DDR factors show increased SUMO2/3 modification in response to SENP6 siRNA depletion which does not always promote increased proteasome mediated turnover. Loss of SENP6 promotes re-localisation of several DDR factors to PML-NBs. However, loss of PML does not reverse this process. Using ERRC1 as a model substrate they show that this relocalisation is driven by SIM dependent interactions, suggesting loss of SUMO homeostasis drives DDR factors into condensates.

The work presented here is of high quality and interest to the field. The model proposed is plausible but there are some limitations to the study that make some of the conclusions superficial. With additional experiments this work would be suitable for publication in Nature Communications. My main concerns/suggestions are highlighted below, and more minor comments listed later.

1. Most experiments are performed in untreated cells, suggesting the spontaneous DNA damage detected here is a response to replication stress. Are SENP6 depleted cells undergoing increased replication fork collapse and processing to DSBs?
2. Further proof that the DDR factors tested here are direct SENP6 substrates is needed, siSENP6 could have indirect effects on the SUMOylation of DDR factors. It would also be useful to demonstrate the changes in SUMOylation DDR factors treated with DSB inducing agents. Does DNA damage further increase the SUMOylation of these factors on SENP6 loss? Does DNA damage alter the interaction of SENP6 with its substrates?
3. Mechanistically the authors do not show that the relocalisation of DDR factors to condensates has any biological relevance to DNA repair. The authors need to show that DDR factors trapped in these condensates are less available for DSB recruitment on SENP6 loss.

Introduction

- "Group modification by SUMO is particularly important for DNA repair (Psakhye and Jentsch, 2012)" – this should be modified to group modification by SUMO is particularly important for DNA repair in yeast as the cited study was carried out in yeast and the functional data for DSB repair group modification in multicellular eukaryotes has not yet been shown conclusively.
- RPA should be corrected to RPA70

Figure 1.

- 1B In addition to Dou et al 2010 and Wagner et al 2019, Schick et al. 2022 also demonstrated increased spontaneous gH2AX formation in Senp6 depleted cells.
- As these gH2AX are occurring in unstressed cells it would be useful to show the cell cycle stage here with an S phase marker such as EdU. I would assume the majority of the DSBs formed on siSENP6 are a result of replication stress.
- If the gH2AX foci are enriched in S phase cells it would be informative to include a MUS81 siRNA co-depletion with SENP6 as this would demonstrate replication associated DSBs are forming on loss of SENP6.
- To confirm the increased gH2AX foci are a result of loss of SENP6 catalytic activity these assays should also include rescues with WT and catalytic deficient SENP6.
- 1E SUMO2/3 +/- gH2AX foci should be shown in the same panel for the NTP as well as the siSENP6 for comparison.
- 1G – Survival curves are usually shown with a log scale. The sensitivity to these DSB inducing agents on SENP6 depletion is surprisingly mild. This may be due to this being a viability assay rather than a colony survival assay. I would recommend repeating this assay with the more sensitive colony

survival analysis.

- 1H – the fixing time after IR is not stated – the failure to detect SENP6 co-localising with gH2AX could be for kinetic reasons. 2.5 Gy is also a rather low, it is also possible that a greater dose would promote co-localisation.
- 1H – including a subcellular fractionation (cytoplasmic, soluble nuclear and chromatin +/- IR) would make the conclusions here stronger as currently the only evidence that SENP6 does not re-localise after IR is from just 3 cells per condition.

Figure 2.

- The proteomic data in figure 2 is interesting but has already been published elsewhere (Liebelt et al 2019 Nat Comms) so does not offer any additional insight. The same experimental set up but in cells treated with a DNA damaging agent would have provided more novel understanding.

Figure 3.

- There is no exogenous DSB induction - suggesting that SENP6 regulates the basal levels of SUMOylation of these factors. At this point, this does not demonstrate that SENP6 plays any role in the DSB repair function of these factors as it does not show regulation of the SUMOylation status of these factors during DDR signalling. SUMOylation of many of these factors has also not been directly shown to modulate their roles in DDR.
- To prove that these factors are direct substrates of SENP6 co-immunoprecipitations should be shown. The catalytic mutant of SENP6 may need to be included as a substrate trap owing to the fast on/off rates of SENPs. Inclusion of a DSB inducing agent such as IR in these Co-IPs may also show dissociation during DDR.
- The data presented does not directly indicate the tested factors are being polySUMOylated as they could also be multi-monoSUMOylated. It is possible that like RNF4, which can detect multiple single SUMO moieties that are spatially close, SENP6 may also deSUMOylate extensively multi-monoSUMOylated factors. A useful control for this would be to repeat these pulldowns with the SUMO2 K0 (lysine-less) mutant that was previously used by the authors group against multi-SUMOylated c-myc (Gonzalez-Prieto 2015). The differences in multi-mono v. polySUMOylation may explain the different outcomes for some of these factors on SENP6 loss.

Figure 4.

- Figure 4 needs loading controls, and preferably quantification (Ponceaus-S isn't sensitive enough to show even loading).

Figure 5

- As SENP6 depletion has considerable impact on cell cycle progression, is the increase in PML/DDR factor localisation due to an enrichment of a specific cell cycle stage? Do these co-localisations occur in all stages of the cell cycle?
- Why do some factors co-localise with PML while others do not? What determines the difference?
- Most of the DDR factors analysed (BRCA1, 53BP1, RAP80, MDC1, ERRC1 and XPF) form DSB associated foci with exogenous DNA damage – do they do this in siSENP6 cells? Does the re-localisation to PML-NBs in undamaged cells prevent or reduce the relocalisation to DSBs?

Figure 6

- Does SENP6 co-localisation with PML-NBs require SENP6 SIM motifs (characterised by Wagner et al 2019). Does SENP6 residency in the PML-NBs promote deSUMOylation and eviction of DDR factors, with loss of SENP6 causing retention in PML-NB bodies?
- What is the localisation pattern on SENP6 in the PML KO clones?

Reviewer #2:

Remarks to the Author:

Summary of manuscript

In this manuscript, Claessens and Vertegaal investigate the role of the poly-SUMO2/3 deconjugating enzyme SENP6 in genome stability. The authors suggest that the major function of SENP6 isn't directly at DNA lesions, but rather within the nucleoplasm, where it limits polySUMOylation of a number of factors involved in the DDR. They then suggest that the hyper-SUMOylation of a range of DDR factors doesn't play a major role in their ubiquitin-dependent degradation, as depletion of SENP6 leads to reduced polySUMOylation of some DDR factors. Instead, the authors claim that a number of polySUMOylated DDR factors, but not all factors examined, become enriched at PML bodies, which show a dramatic increase in number after SENP6 depletion. The authors suggest that the presence of SUMO-interacting motifs in DDR factors promotes their localisation to PML bodies, rather than merely their SUMOylation status, using ERCC1 as an example of this. Surprisingly, genetic abrogation of PML had no impact on the ability of various DDR factors to form foci after SENP6 depletion, with the authors suggesting that hyper SUMOylation of these factors is sufficient to promote formation of protein condensates. Overall, hyper polySUMOylation of DDR factors in the absence of SENP6 is proposed to drive SUMO-SIM interactions in both a PML-dependent and independent manner, which prevents recruitment to DNA lesions, impairing DNA repair and driving genome instability.

Strengths:

- Convincing cellular phenotypes and cellular biochemistry, with data generally well controlled (see some exceptions below)
- Findings that will be of interest to a wide audience across DNA repair and ubiquitin/SUMO/UBL fields.
- Proposes a degradation-independent signalling function for SUMOylation in the regulation of DDR factors.

Weaknesses:

- Some data is repetitious with previous publications from the Vertegaal lab
- Lack of mechanism for foci formation after PML loss
- Suggested model is not entirely supported by the data
- Model is not universally applicable, yet no discussion is made as to why some DDR factors might be impacted and others not.
- SUMO-SIM model already a well known mechanism of PML body formation

Overall:

Due to the major criticisms noted below, it would be difficult to support the manuscript in its current seven figure format, though, with additional controls and experiments to strengthen the current data, I could potentially support a shorter article format.

Major points:

Point 1.

Figures 1-3 are essentially data that has already been published (except Fig 1C), so there's limited novelty here and the majority could go in the Supplementary. I'd suggest that these should be consolidated and the article made into a shorter format.

Point 2.

The suggestion that SENP6 functions within the nucleoplasm is based on the authors inability to detect SENP6 at IR-induced γ H2AX foci. However, given that DNA triggers PIAS-dependent SUMOylation of DDR factors (Psakhye and Jentsch, 2013), why would the deSUMOylation machinery act predominantly within the nucleoplasm as the authors claim?

Furthermore, without performing similar localisation studies as previous studies, it seems rather pointless to make comparisons. Did the authors use biochemical fractionation for SENP6 localisation after DNA damage? Or did they try the PLA assay?

Point 3.

The authors state that "non-modified protein levels of CtIP, BLM and BRCA1 decreased, indicating that a buildup of SUMO2/3 polymers on these proteins could have led to their degradation or alternatively to such extensive SUMOylation that the amount of non-modified proteins was reduced". It is essential to provide more evidence to address this. To distinguish between the two, I suggest the authors take recombinant SENP6 and add it to the cell lysate or pull-down, then blot back for CtIP, BLM, BRCA1 etc... At present, there's no reason proposed as to why polySUMOylation of these factors would be reduced after SENP6 depletion. A review from the Cohen lab explains how polyubiquitylation can lead to spurious conclusions derived from immunoblotting when the substrate is heavily modified by ubiquitin, which is analogous to the data presented here:
<https://www.ncbi.nlm.nih.gov/pmc/articles/PMC4709362/>

Furthermore, the authors claim that "Taken together, these data rule out that SUMO2/3 polymers on these proteins are substrates for STUbLs and facilitate degradation." To test this more rigorously, the authors would have to assay protein turnover of these proteins over a timecourse.

Point 4.

A major weakness of the manuscript is the lack of mechanism for foci formation in the absence of PML. To start to understand the mechanism of foci formation in the absence of PML (Figure 7), the authors could at least use the SUMO E1i to see if SUMOylation is still required for foci formation of ERCC1, XPF, BRCA1, BLM and 53BP1 (similar to the experimental set-up in Figure 5). Presumably it will, so the authors can then use the ERCC1 WT vs SIM mutant to understand if the SUMO-SIM model still applies in the absence of PML. In addition to this, it would be useful to characterise further the foci formed in the absence of PML and SENP6 – can the authors take a panel of known PML-associated factors and assess whether the PML-independent foci formation of the DDR factors still contains canonical PML-associated factors?

Point 5.

U2OS cells use the alternative lengthening of telomeres pathway, which are microscopically discernible and to which PML localises. The authors should determine whether the factors they have examined here are mainly localised to telomeres. They could approach this using RAP1 or some other factor found in telomeres. Moreover, they should take a cell line(s) that doesn't use ALT and repeat their key experiments.

Point 6.

The authors propose in their model that the hyper polySUMOylation of DDR factors leads to their accumulation in PML and SUMO condensates, which inhibits their ability to localise to DNA damage sites. However, there is no data in the manuscript which shows that the DDR factors described fail to properly localise to genotoxin-induced DNA lesions in the absence of SENP6 – quantification +/- DNA damage would be required.

Point 7.

Why is median number of foci often used throughout the manuscript?

Minor points:

Figure 1:

- B - SUMO1 would be a useful control here
- E - Is this useful without a control?
- F - It is not clear how many cells were quantified to assess micronuclei formation following siSENP6.
- G - SENP6 depletion is lethal, as illustrated by DepMap and the authors own previous clonogenic data. Thus, presumably there is still some level of SENP6 in the cells after 5 days. This caveat should be stated in the text.

Figure 3:

- A - It would have been useful to rescue the stable shRNA SENP6 cells with WT and catalytically inactive SENP6 to test if the protease activity is responsible. A SUMO1 blot should also be included here.

Figure 4:

- It would be better to place the inputs and pulldown for each target together, rather than split them across different panels.

Figure 5:

- It would be important to split the green and purple channels, as well as having the merged image.

Figure 6:

- The ERCC1-GFP(SIM) mutant seems to be much more cytoplasmic than WT – a biochemical fractionation would illustrate its distribution. Is there sufficient nuclear localisation to assess its recruitment to foci?
- Are the SUMOylation sites of ERCC1 well defined? It might be useful to introduce K>R mutants to complement the SIM mutant data.
- D - Running the input and IP samples on the same gel would enable improved comparison to the input levels.
- H - The immunoblots for the quantification of co-purified XPF could be shown in parallel.

Figure 7:

- A - Higher magnification images may be useful to demonstrate the localisation of PML and γ H2AX foci in response to SENP6 knockdown.

Reviewer #3:

Remarks to the Author:

The manuscript by Claessens and Vertegaal extends previous work by the Vertegaal laboratory (Mitra et al., Nature communications, 2019) and other research groups (Gibbs-Seymour et al., Mol Cell, 2015, Dou et al., Mol Cell, 2010, Wagner et al., Cell reports 2019) on the importance of SENP6 for the maintenance of genomic stability. As such, the manuscript is clearly written, the experiments are of high quality and data are presented with proper controls. However, some results (large parts of Figure 1 and Figure 2) are just a reiteration of already available/published data and therefore provide little novelty. Along the same line, Figure 3 is solely a validation of the previously published mass spectrometry (Mitra et al., Nature communications, 2019). Furthermore, it was previously described that SENP6 depletion leads to retention of SUMO chains on a subset of DDR factors leading to reduced chromatin recruitment (Schick et al., 2022). The most interesting novel aspect provided here is the observation that at least for a subset of repair factors this is due to their hyperaccumulation in nuclear condensates, such as PML speckles. This is in line with the described role of SENP6 as a regulator of PML body (Hattersley et al., MCB 2011, Mukhopadaya et al., JCB, 2006) and strengthens the concept that SUMO-SIM interactions act as a scaffold of PML body and other nuclear SUMO condensates (Banani et al., Cell, 2016). Altogether, the manuscript provides a mechanistic explanation how knock-down of SENP6 reduces the presence of DDR factors at sites of DNA damage. The authors propose that supraphysiological levels of polySUMO chain induce the trapping of DDR factors in nuclear condensates. While this is a potentially interesting finding the conceptual novelty of the work is somewhat limited. Further, some aspects of the work need to be clarified or strengthened.

Major points:

1) Data presented here support the idea that polySUMO chains exhibit important regulatory functions beyond their function as recruitment platforms for the SUMO targeted ubiquitin ligase RNF4. However, I do feel that this aspects needs to be solidified, because previous work has shown that

polySUMOylation of DDR factors result in their interaction with RNF4 and subsequent degradation after DNA damage (for example Galanty et al., *Genes and Dev*, 2012, Garvin et al., 2019). Further, RNF4 is known to reside at PML body possibly indicating that trapping in condensates precedes degradation. Of note, previous work has shown that ablating the expression of SENP6 also leads to ubiquitin-mediated proteolysis of RNF4. Thus, it is conceivable that the DDR factors investigated here might represent StUbl substrates under physiological conditions (e.g. regulated acute inactivation of SENP6), but are not degraded in the experimental system used here due to the constitutive inactivation of SENP6 followed by loss of RNF4. Therefore, RNF4 levels in SENP6 depleted cells should be controlled. Further, the fate of the polySUMOylated DDR factors upon re-expression of RNF4 should be monitored. To circumvent the drawback of constitutive SENP6 inactivation by siRNA the authors may also consider induced degradation of SENP6 protein, which may better reflect the acute regulation of SENP6 (Dou et al., *Mol Cell*, 2010).

2) The authors conclude that, in the absence of SENP6, supraphysiological levels of polySUMO chains induce the trapping of DDR factors in nuclear condensates thereby reducing their presence at sites of DNA damage. At this stage, however, the physiological relevance and regulation of this process remains elusive. Is the association of SENP6 with the respective DDR factors induced upon DNA damage in order to trigger their recruitment to sites of DNA damage? Is the activity of SENP6 altered in response to genotoxic insults?

3) The authors conclude that accumulation of polySUMO chains upon depletion of SENP6 traps DDR factors in nuclear condensates. To strengthen the conclusion that this a polySUMO and SENP6 dependent process other SENP proteins should be tested as well. Further, the role of monoSUMOylation vs. polySUMOylation should be explored.

Additional points:

3. In Figure 3, the input lanes for BLM and CtIP seems to have less respective protein levels in response to SENP6 depletion. At least for BLM, higher molecular weight regions also did not show any smear/bands. So, the claim that the decrease of the protein levels of BLM and CtIP can partially be because of reduction of SUMO-unmodified forms of the proteins is not justifiable.
5. The chain specificity of Ub (K48, K63) in Figure 4D is worth exploring in the current context.
6. Figure 6C, D: ERCC1 SIM mutants seem to have more expression/stability in cells. Can the authors provide some explanations for this?
7. As such, Figure 6H is not relevant to the observations presented here and can go in the supplementary section. ERCC1 SIM domain rather than ERCC1 SUMOylation seems to be important for localizing to PML bodies and for interaction with XPF. However, ML792 treatment sensitizes ERCC1-XPF1 interaction. Does SUMOylated XPF1 binds to ERCC1 by the SIM domain of the latter?
8. In PML KO cell line, do foci for all these DDR proteins co-localize? As ERCC1-XPF is shown to be SUMO-SIM dependent, at least these two proteins should co-localize. Are any non-canonical foci formed in absence of PML where some of these polySUMOylated proteins co-localize? If so, does ML792 treatment dissolve these foci (not shown in Figure 5D)?

REPLY TO REVIEWERS

We thank the reviewers for the critical appraisal of our manuscript. Because of the large number of comments and some critical changes to our original model, we summarize below the major aspects that are altered or added in the revised manuscript, before providing a point-by-point response to each comment.

1. We prioritized experiments that were critical for revising the model. Since we could not causally relate the accumulation of DDR proteins in (PML) condensates to a defect in DDR signalling at DNA damage sites, we focused on experiments relating to DDR signalling at the DNA damage sites in SENP6-depleted cells and the observed genomic defects, rather than further exploring or solidifying our findings on condensation of the DDR proteins in (PML) nuclear condensates.
2. We removed data from the SUMO2 pulldown experiment in SENP6 shRNA transduced cells treated with MG132. Currently, we cannot formally exclude a potential role for SENP6 and RNF4-mediated proteasomal degradation in regulating the identified DDR targets. The absence of an effect from MG132 treatment on SUMOylation levels, in contrast to RNF4 co-depletion, could perhaps result from more severe exhaustion of the SUMO and ubiquitin pool with MG132 treatment or different timing of the MG132 treatment and RNF4 knockdown in relation to the SENP6 knockdown.
3. We performed experiments looking at **SUMO2/3 kinetics** at **IR-induced DNA damage foci** and **UVA-laser tracks**. Knocking down SENP6 resulted in an excess of SUMO2/3 at DNA damage sites that was not cleared in a timely fashion. We ruled out that this effect could be ascribed to the loss of RNF4, by including siRNF4 cells in the IR experiment. We observed similar kinetics for BRCA1 and 53BP1. The observed kinetics argue against our original mechanistic model in which supraphysiological levels of SUMOylation traps SUMOylated proteins in (PML) nuclear condensates, causing genomic instability as a result of DNA repair proteins not being recruited to DNA damage sites. The genomic instability and accumulation of DNA repair proteins in (PML) nuclear condensates in siSENP6 cells appear to be independent phenotypes that result from a disbalance in SUMOylation levels of these proteins and are not causally linked to one another.
4. We performed **SUMO2 pulldown experiments** in a cell line expressing **lysine-deficient His10-SUMO2** (His10-SUMO2 K0) and found that SENP6 mainly counteracts **poly-SUMOylation** of DNA repair proteins under otherwise unstressed conditions rather than **multi-mono-SUMOylation** (validated proteins: BRCA1, BARD1, BLM and 53BP1).
5. We performed **SUMO2 pulldown experiments** in the presence of a genotoxic agent, **hydroxyurea**, and found that SENP6 does not only maintain DNA repair proteins in a hypo-SUMOylated state under otherwise unstressed conditions, but also counteracts their SUMOylation during active DNA damage signalling (validated proteins: BRCA1, BARD1, MDC1 and BLM).
6. We performed **SUMO2 pulldown experiments** with **SENP6 and RNF4 single- and double-knockdown** to gain more insight into the potential antagonizing role of SENP6 and RNF4 in regulating the SUMOylation levels of DNA repair proteins. Knockdown of RNF4 and SENP6 led to a further increase in total SUMO conjugation levels and SUMO conjugation levels of BLM, BRCA1 and BARD1, compared to the single knockdowns.
7. We performed **control western blots for RNF4** to determine RNF4 levels under our experimental conditions. The build-up of SUMO2/3 chains in our SUMO2 pulldown experiments with SENP6 shRNA can only be ascribed to the depletion of SENP6 as RNF4 levels remained unaffected. However, in our experiments with siSENP6 cells (both 72h and 96h knockdown) RNF4 gets activated and degraded, consistent with what has been described in the literature.
8. We performed **γH2AX and SUMO2/3 immunofluorescence** in **siSENP6** and **siRNF4** cells and found that the spontaneous induction of γH2AX foci in siSENP6 cells cannot be ascribed to the loss of RNF4. The depletion of only RNF4 also resulted in a significantly more modest increase in SUMO2/3 foci than depletion of SENP6 did.
9. We performed additional **immunofluorescence experiments** looking at **SENP6 localization at IR-induced DNA damage foci**, by including extra timepoints, a higher dosage and

performing a pre-extraction step, as well as its localization at DNA damage sites using **UVA-laser micro-irradiation**. Exogenously expressed **wildtype and catalytically inactive GFP-SENP6** were rarely observed at the UVA-laser induced DNA damage tracks, but never at IR-induced DNA damage foci. We were not able to detect accumulation of the endogenous protein at any site of DNA damage.

We have revised our model accordingly: SENP6 counteracts polySUMOylation of DDR proteins under unstressed conditions and in response to genotoxic stress. Depletion of SENP6 leads to supraphysiological levels of SUMOylation, activating the RNF4 pathway. Functionally, this results in uncoordinated recruitment and persistence of SUMOylated proteins at DNA damage sites, as well as their accumulation in (PML) nuclear condensates, in a SUMO-conjugation and SIM-dependent manner.

Reviewer #1 (Remarks to the Author):

Claessens & Vertegaal build on previous work that identify the SUMO protease SENP6 as an important regulator of various groups of proteins including factors of the DDR network. They show that spontaneous DSB repair signalling is increased in SENP6 depleted cells, resulting a surprisingly mild sensitivity to DSB inducing agents. They confirm several important DDR factors show increased SUMO2/3 modification in response to SENP6 siRNA depletion which does not always promote increased proteasome mediated turnover. Loss of SENP6 promotes re-localisation of several DDR factors to PML-NBs. However, loss of PML does not reverse this process. Using ERRC1 as a model substrate they show that this relocalisation is driven by SIM dependent interactions, suggesting loss of SUMO homeostasis drives DDR factors into condensates.

The work presented here is of high quality and interest to the field. The model proposed is plausible but there are some limitations to the study that make some of the conclusions superficial. With additional experiments this work would be suitable for publication in Nature Communications.

My main concerns/suggestions are highlighted below, and more minor comments listed later.

We thank the reviewer for support. We appreciate the extensive critical appraisal and suggestions for improvement and solidification of the proposed model.

1. Most experiments are performed in untreated cells, suggesting the spontaneous DNA damage detected here is a response to replication stress. Are SENP6 depleted cells undergoing increased replication fork collapse and processing to DSBs?

We agree that replication stress could be one of the initiating events leading to spontaneous DNA damage after siRNA-knockdown of SENP6. Multiple papers point towards a role for SENP6 in replication stress (DOI: [10.1016/j.molcel.2010.07.021](https://doi.org/10.1016/j.molcel.2010.07.021) and [10.1016/j.celrep.2019.08.106](https://doi.org/10.1016/j.celrep.2019.08.106)). These findings align with our cell viability assays (Figure 1F in the revised manuscript), showing increased sensitivity in siSENP6 knockdown cells to reagents causing replication fork stalling and collapse (CPT, MMC, HU). Further exploring replication stress functionally by DNA fibre assays or other assays looking directly at replication fork collapse and processing to DSBs is potentially very interesting in the context of SENP6 and its target proteins, but beyond the scope of this paper and will not provide substantial novelty. Additionally, based on current data from our lab and others, SENP6 is not exclusively involved in replication stress-related damage and has potential roles in other types of DNA damage repair as well.

2. Further proof that the DDR factors tested here are direct SENP6 substrates is needed, siSENP6 could have indirect effects on the SUMOylation of DDR factors. It would also be useful to demonstrate the changes in SUMOylation DDR factors treated with DSB inducing agents. Does DNA damage further increase the SUMOylation of these factors on SENP6 loss?

The reviewer raises an interesting point about the SUMOylation levels of the identified SENP6 substrates in the presence of DNA damage. For multiple proteins in our dataset, increased SUMOylation has been shown before by mass-spectrometry or western blotting upon different DNA damage treatments. Examples are MDC1, 53BP1 and BLM upon IR and/or UV treatment (DOI: [10.1093/nar/gkz977](https://doi.org/10.1093/nar/gkz977)); BRCA1, BARD1, XPF, USP28 and UIMC1 upon MMC treatment in G1-S synchronized cells (DOI: [10.1016/j.celrep.2017.09.059](https://doi.org/10.1016/j.celrep.2017.09.059)); BARD1 upon high dose IR, MMS and Bleocin treatment in the presence of MG132 (DOI: [10.1038/s41467-017-01900-x](https://doi.org/10.1038/s41467-017-01900-x)); BLM upon HU treatment (DOI: [10.3389/fgene.2021.753535](https://doi.org/10.3389/fgene.2021.753535)); BRCA1, BARD1, BLM and MDC1 upon HU treatment (DOI: [10.1074/mcp.O114.044792](https://doi.org/10.1074/mcp.O114.044792)). For proof-of-concept and because of a confirmed role for SENP6 in replication stress-related genomic instability, we looked at SUMOylation levels of BRCA1, BARD1, BLM and MDC1 with 2 mM HU treatment for 24 hours in control shRNA transduced

cells and SENP6 shRNA transduced cells. Albeit subtle and variable, we observed increases in the SUMOylation levels of these proteins compared to HU treatment or SENP6 knockdown alone (A). We quantified the fold change of the SUMOylation levels of the DNA repair proteins across three biological replicates by dividing the intensity measured for the SENP6 knockdown cells treated with HU by the intensity for the DMSO-treated SENP6 knockdown cells (B). Because of the substantial increase in SUMOylation levels of some proteins in the single conditions, we speculate that the margin for detecting further increases is limited. We have included these data as Figure 4 in the revised manuscript.

Does DNA damage alter the interaction of SENP6 with its substrates?

Some validated SENP6 substrates have been confirmed by their physical interaction under specific conditions (RPA70 DOI: [10.1016/j.molcel.2010.07.021](https://doi.org/10.1016/j.molcel.2010.07.021); EXO1 DOI: [10.1080/15384101.2015.1060381](https://doi.org/10.1080/15384101.2015.1060381); FANCI DOI: [10.1016/j.molcel.2014.12.001](https://doi.org/10.1016/j.molcel.2014.12.001); PDS5B DOI: [10.1016/j.celrep.2019.08.106](https://doi.org/10.1016/j.celrep.2019.08.106)). Wagner *et al.* (DOI: [10.1016/j.celrep.2019.08.106](https://doi.org/10.1016/j.celrep.2019.08.106)) performed an unbiased mass-spectrometry screen by immunoprecipitation of cells transiently expressing catalytically inactive Flag-SENP6. Our validated SENP6 targets were either not present in their dataset or identified but not as a significant interactor. As this was under untreated conditions in unsynchronized cells, we cannot rule out that SENP6 interacts with our identified targets in the presence of DNA damage and/or during a specific cell cycle phase. Exploring this properly would require testing a broad range of DNA damage treatments, with different doses,

time courses and potentially cells in specific cell-cycle phases, using a mass-spectrometry-based approach. Nonetheless, we respectfully disagree that the absence or presence of a physical interaction between SENP6 and the unmodified proteins yields a conclusive answer to whether these proteins are direct substrates, as it could be that only the heavily SUMO-modified protein fraction interacts with SENP6, which is difficult to detect endogenously.

3. Mechanistically the authors do not show that the relocalisation of DDR factors to condensates has any biological relevance to DNA repair. The authors need to show that DDR factors trapped in these condensates are less available for DSB recruitment on SENP6 loss.

We agree with the reviewer that this part of our mechanistic model was still speculative and required experiments directly addressing causality between the observed genomic instability and localization of DNA repair proteins in (PML) condensates. We have addressed this with two independent approaches: determining the presence of SUMO2/3 at DNA damage tracks induced with UVA-laser microirradiation and at DNA damage foci induced with gamma-irradiation in siSENP6 cells.

We quantified the amount of SUMO2/3 at UVA-laser-induced DNA damage sites 72 hours post-transfection in siSENP6 and mock-transfected (NTP) cells (A). Cells were pre-sensitized with 10 μ M BrdU for 24 hours. Laser-tracks were made with two laser powers, 7% and 10%, and cells were pre-extracted and fixed after 1 hour. Cells were subsequently stained with γ H2AX and SUMO2/3. γ H2AX and SUMO2/3 intensity were quantified by quantifying the intensity at the laser-track and subtracting it with the intensity in the nucleoplasm, to determine the amount of γ H2AX and SUMO2/3 that specifically accumulated at the tracks. The γ H2AX intensity is similar in mock-transfected and siSENP6-transfected cells, while SUMO2/3 intensity at the tracks is higher for both laser-powers used in siSENP6 cells (B). We have added these data to Figure 6A and supplementary Figure 4D in the revised manuscript.

We repeated this experiment with additional timepoints, 1h, 2h, 4h and 8h, and observed more SUMO2/3 accumulation in the nucleus and specifically at the UVA-induced laser-tracks at all timepoints (A). Knockdown of SENP6 was confirmed by immunoblotting (B). We have added these data to Figure 6B and Supplementary Figure 4E in the revised manuscript.

In the second approach, we also observed more SUMO2/3 foci and colocalization with γ H2AX at IR-induced damage foci in siSENP6 cells 72 hours post-transfection compared to mock-transfected cells (NTP). γ H2AX foci were induced by 2.5 Gy irradiation, followed by measuring γ H2AX and SUMO2/3 foci at 1 min, 1h, 4h and 8h post-IR treatment. The absolute number of (colocalizing) foci per nucleus was quantified (A), as well as the percentage of cells with more than 5 SUMO2/3 foci. γ H2AX and BRCA1 (B) and γ H2AX and 53BP1 (C) co-staining in the same experiment revealed more BRCA1 foci and colocalization with γ H2AX at early timepoints, and more 53BP1 foci and colocalization with γ H2AX at all timepoints. Knockdown of SENP6 was confirmed by immunoblotting (D). We have added these data to Figure 6C-H and Supplementary Figure 4F in the revised manuscript.

Collectively, this would suggest that there is not a lack of SUMO2/3 or SUMO2/3-modified DDR proteins at DNA damage sites in siSENP6 knockdown cells, but actually an excess, arguing against a model in which the SUMOylated DDR proteins are trapped in (PML-) condensates and cannot localize to these sites. We have revised our model accordingly.

We are also aware of the RNF4 depletion in siSENP6 knockdown cells and a potential combinatorial effect of co-depleting both proteins in these functional DNA damage experiments. We therefore compared siSENP6 knockdown cells with siRNF4 knockdown cells and performed immunofluorescence for γH2AX and SUMO2/3. There was a significant increase in spontaneous γH2AX foci formation in siSENP6 knockdown cells compared to mock-transfected (NTP) cells (as shown before by others and confirmed by us in Figure 1B-D of the manuscript) and siRNF4 cells (A). There was no increase in γH2AX foci in siRNF4 cells compared to NTP cells, suggesting that the increase in γH2AX foci in siSENP6 cells is not only due to the degradation of RNF4. There is a significant increase in SUMO2/3 foci in siRNF4 cells compared to NTP cells, and a significant increase in SUMO2/3 foci in siSENP6 cells compared to both NTP and siRNF4 cells (A). This suggests that also the increase in SUMO2/3 foci in siSENP6 cells is not only due to the degradation of RNF4. Knockdown of both proteins was confirmed by immunoblotting (B). We also validated the SUMO2/3 kinetics observed after gamma-irradiation described above by comparing siSENP6 with siRNF4 cells (C). Similarly, in the presence of an exogenous source of DNA damage, the increase

in SUMO2/3 foci is not only due to the degradation of RNF4, but also requires depletion of SENP6. We have included these data in Supplementary Figure 5B-D in the revised manuscript.

RNF4 levels are also discussed in more detail in major comment #1 from reviewer #3.

Introduction

● "Group modification by SUMO is particularly important for DNA repair (Psakhye and Jentsch, 2012)" – this should be modified to group modification by SUMO is particularly important for DNA repair in yeast as the cited study was carried out in yeast and the functional data for DSB repair group modification in multicellular eukaryotes has not yet been shown conclusively.

We have added the specification for yeast.

● RPA should be corrected to RPA70

We have made this correction.

Figure 1.

● 1B In addition to Dou et al 2010 and Wagner et al 2019, Schick et al. 2022 also demonstrated increased spontaneous γH2AX formation in Senp6 depleted cells.

We have added this reference.

● As these γH2AX are occurring in unstressed cells it would be useful to show the cell cycle stage here with an S phase marker such as EdU. I would assume the majority of the DSBs formed on siSENP6 are a result of replication stress.

We again agree with the reviewer raising the point of replication stress in siSENP6 cells and that γ H2AX foci formation could be mainly triggered during S-phase in these cells. In the context of cell cycle progression, Wagner *et al.* (DOI: [10.1016/j.celrep.2019.08.106](https://doi.org/10.1016/j.celrep.2019.08.106)) demonstrate impaired checkpoint signaling after aphidicolin-induced replication stress, suggesting that cells initially still progress through the cell cycle despite the presence of unresolved replication stress. Liebelt *et al.* (DOI: [10.1038/s41467-019-11773-x](https://doi.org/10.1038/s41467-019-11773-x)) observed an increased amount of cells in G2/M phases and a concomitant decrease in cells in G1 with shRNA-mediated knockdown of SENP6 as determined by propidium iodide staining and flow cytometry; the amount of cells in S-phase remained unchanged. Furthermore, Dou *et al.* (DOI: [10.1016/j.molcel.2010.07.021](https://doi.org/10.1016/j.molcel.2010.07.021)) showed a delay in S phase and accumulation of cells in S and G2/M phases after cell synchronization, as well as a decrease in BrdU incorporation, in siSENP6 cells. Collectively, these data already illustrate the importance of SENP6 in preventing replication and cell cycle defects.

• **If the γ H2AX foci are enriched in S phase cells it would be informative to include a MUS81 siRNA co-depletion with SENP6 as this would demonstrate replication associated DSBs are forming on loss of SENP6.**

We agree with the reviewer that this could be an informative experiment to further expand on replication stress, but we believe that exploring this is beyond the scope of our manuscript.

• **To confirm the increased γ H2AX foci are a result of loss of SENP6 catalytic activity these assays should also include rescues with WT and catalytic deficient SENP6.**

We initially planned to address this with shRNA-mediated knockdown of SENP6, as we have done similar experiments before looking at the fate of centromeric proteins, utilizing cells expressing shRNA-resistant wildtype GFP-SENP6 and catalytically inactive SENP6. However, with shRNA-mediated knockdown of SENP6 we do not observe a considerable increase in γ H2AX foci formation, presumably because of a less efficient knockdown, as there was also no clear increase in SUMO2/3 foci (A-B). Increasing the time to 96 hours post-transduction, did not enhance the phenotype (C). Thus, the shRNA approach was not suitable for addressing this point.

• **1E SUMO2/3 -/+ γ H2AX foci should be shown in the same panel for the NTP as well as the siSENP6 for comparison.**

We have removed this figure from the revised manuscript.

• **1G – Survival curves are usually shown with a log scale. The sensitivity to these DSB inducing agents on SENP6 depletion is surprisingly mild. This may be due to this being a**

viability assay rather than a colony survival assay. I would recommend repeating this assay with the more sensitive colony survival analysis.

We would like to point out that the data are presented on a log-scale but that for visual purposes we had set the start of the y-axis at 10. We initially performed colony survival assays with shRNA-mediated knockdown of SENP6, re-seeding the cells at limiting density, followed by IR treatment. However, this did not give conclusive results. The knockdown of SENP6 was already very detrimental to the cells over the time-course of the colony survival assay (10-14 days), perhaps not providing a window for an additional effect of the irradiation. We also observed that the cells that did grow out in colonies at the end of the colony survival assay (re-)expressed SENP6. We switched to cell viability assays, because shorter time-courses can be used till measurement after the depletion of SENP6. Furthermore, Garvin *et al.* (doi: [10.1101/gad.321125.118](https://doi.org/10.1101/gad.321125.118)) have already shown reduced colony survival in siSENP6 cells treated with IR and re-seeded for colony outgrowth.

• 1H – the fixing time after IR is not stated – the failure to detect SENP6 co-localising with γ H2AX could be for kinetic reasons. 2.5 Gy is also a rather low, it is also possible that a greater dose would promote co-localisation.

We thank the reviewer for pointing out that this information is lacking and have included this in the revised manuscript; cells were fixed after 1 hour. We have moved these data to Supplementary Figure 1F of the revised manuscript. We cannot completely rule out that the failure to detect SENP6 at IR-induced γ H2AX foci is because we are missing the correct timepoint or dose. We therefore repeated the experiment with 5 Gy and included multiple timepoints (10min, 1h and 4h) and also used a CSK pre-extraction method to enrich for chromatin-associated SENP6, but failed to observe accumulation of SENP6 at the γ H2AX foci; SENP6 expression remained diffuse throughout. This was the case for the endogenous protein, but also for wildtype and catalytically inactive GFP-SENP6 expressed in cells. Representative micrographs are shown from three biological replicates. We have included these data as Supplementary Figure 2 in the revised manuscript.

Gibbs-Seymour *et al.* (doi: [10.1016/j.molcel.2014.12.001](https://doi.org/10.1016/j.molcel.2014.12.001)) described localization of exogenously expressed catalytically inactive HA-SENP6 at FANCI foci after MMC treatment, but failed to observe localization at laser-irradiated sites. Since SENP6 is also involved in regulation of other cellular processes and is not exclusively or predominantly a DNA damage response protein, it is perhaps not unexpected that SENP6 expression and localization remains diffuse despite the presence of DNA damage. Moreover, cellular fractionation assays from Wagner *et al.* (DOI: [10.1016/j.celrep.2019.08.106](https://doi.org/10.1016/j.celrep.2019.08.106)) show SENP6 is present in both the chromatin and nucleoplasmic fractions under control conditions, and that aphidicolin treatment did not lead to a re-localisation of SENP6 detectable by western blotting. We also did a similar cellular fractionation assay under control conditions and in cells treated with 4 Gy IR at 5min, 1h and 4h post-treatment. We did not observe altered SENP6 levels in the chromatin-bound fraction for endogenous SENP6. In wildtype GFP-SENP6 expressing cell lines, we observed perhaps slightly higher SENP6 levels in the chromatin fraction at 5 minutes post IR. Also here, we cannot rule out overexpression artefacts.

Sirbu *et al.* (doi: [10.1074/jbc.M113.511337](https://doi.org/10.1074/jbc.M113.511337)) performed iPOND mass-spectrometry to identify proteins at stalled-replication forks and observed recruitment of SENP6 at early timepoints after induction of replication fork stalling (at 10 minutes, but not yet at 5 minutes and no longer at 15 minutes), suggesting that there could be very local and transient enrichment of SENP6 upon replication stress or DNA damage conditions, but that cellular fractionation assays and standard immunofluorescence lack the sensitivity required to reliably and consistently detect this re-localization for the endogenous protein.

● **1H – including a subcellular fractionation (cytoplasmic, soluble nuclear and chromatin -/+ IR) would make the conclusions here stronger as currently the only evidence that SENP6 does not re-localise after IR is from just 3 cells per condition.**

The reviewer's comment that we are basing this on 3 cells per condition is incorrect. In the figure legend we specify: "U2OS cells were exposed to irradiation (IR) at a dose of 2.5 Gy and fixed for microscopy. Cells were stained for Hoechst, γH2AX (green) and SENP6 (magenta). Three images were acquired per biological replicate ranging from 15-25 cells per image (n_{TOTAL} : 173 cells across three biological replicates). Middle and bottom panel: U2OS cells stably expressing wildtype GFP-SENP6 (GFP-SENP6_{WT}) and a catalytic dead GFP-SENP6 (GFP-SENP6_{CD}) under a doxycycline-inducible promoter were induced with doxycycline for 24 hours, exposed to irradiation (IR) at a dose of 2.5 Gy and fixed for microscopy. Cells were stained for Hoechst, γH2AX (green) and GFP (magenta). For GFP-SENP6_{WT} four images and for GFP-SENP6_{CD} five images were acquired per biological replicate consisting of 6-18 GFP⁺ cells per image (n_{TOTAL} =148 and n_{TOTAL} =121 cells across three biological replicates, respectively). Representative images were chosen. Scale bar indicates 10 μm." For visual purposes we have selected inserts of 2-3 cells per condition.

For the discussion of subcellular fractionations, we would like to refer to our reply in the related comment above.

Figure 2.

● **The proteomic data in figure 2 is interesting but has already been published elsewhere (Liebelt et al 2019 Nat Comms) so does not offer any additional insight. The same experimental set up but in cells treated with a DNA damaging agent would have provided more novel understanding.**

We agree with the reviewer that this could have offered additional insights and novelty. Because of the limited numbers of targets in previous SUMO-mass-spectrometry screens for a single treatment condition (type of reagent, dose and timepoint), we used the same experimental set-up as in the original SENP6 proteomic screen, but with 2 mM HU treatment for 24 hours, and validating overlapping proteins from both screens by western blotting, rather than performing new mass-spectrometry screens. These data are described in our reply to comment #2 above.

Figure 3.

● **There is no exogenous DSB induction - suggesting that SENP6 regulates the basal levels of SUMOylation of these factors. At this point, this does not demonstrate that SENP6 plays any role in the DSB repair function of these factors as it does not show regulation of the SUMOylation status of these factors during DDR signalling. SUMOylation of many of these factors has also not been directly shown to modulate their roles in DDR.**

The reviewer is correct that there is no exogenous DNA damage induction in our His10-SUMO2-pulldown experiments where we validate the build-up of SUMO2/3 chains on the DNA damage response proteins in the absence of SENP6. Also, under endogenous conditions, DDR signalling is required to maintain genomic integrity throughout the cell cycle. To get a more conclusive picture of the regulation of the SUMO2/3 levels of these proteins by SENP6 during DDR signalling, we followed the reviewer's earlier suggestions to repeat our His10-SUMO2 pulldown experiments with a DNA damage inducing agent known to increase the SUMOylation levels of some of our identified SENP6 targets. This is described in more detail in comments above.

● **To prove that these factors are direct substrates of SENP6 co-immunoprecipitations should be shown. The catalytic mutant of SENP6 may need to be included as a substrate trap owing to the fast on/off rates of SENPs. Inclusion of a DSB inducing agent such as IR in these Co-IPs may also show dissociation during DDR.**

This comment is related to earlier comments and discussed above.

● **The data presented does not directly indicate the tested factors are being polySUMOylated as they could also be multi-monoSUMOylated. It is possible that like RNF4, which can detect multiple single SUMO moieties that are spatially close, SENP6 may also deSUMOylate extensively multi-monoSUMOylated factors. A useful control for this would be to repeat these pulldowns with the SUMO2 K0 (lysine-less) mutant that was previously used by the authors group against multi-SUMOylated c-myc (Gonzalez-Prieto 2015). The differences in multi-mono v. polySUMOylation may explain the different outcomes for some of these factors on SENP6 loss.**

The reviewer raises an interesting point and although SENP6 is generally accepted to be a SUMO2/3-chain deconjugating enzyme, we can indeed not rule out completely that knockdown of SENP6 might also lead to extensively multi-mono-SUMOylated proteins. To address this, we did His10-pulldown experiments with shRNA-mediated SENP6 knockdown in our His10-SUMO2 wildtype and His10-SUMO2 K0 (lysine-deficient) cell lines and performed western blotting against a selection of our identified SENP6 targets. We predominantly observe polySUMOylation of these proteins in the absence of SENP6 rather than multi-mono-SUMOylation (A). We do not expect exclusively multi-mono-SUMOylation to occur in the His10-SUMO2 K0 cell line, because endogenous SUMO(s) can be conjugated with His10-SUMO2 K0. The His10-SUMO2 K0 levels are somewhat unstable in our experience, leading to minor variations in SUMO2/3 levels between the two cell lines across independent experiments. We therefore quantified the fold change in the

SUMOylation levels of the DNA repair proteins after SENP6 knockdown for two biological replicates by dividing the intensity in His10-SUMO2 wildtype cells by the intensity in His10-SUMO2 KO cells, corrected for the fold change in total SUMO2/3 levels between these cell lines (B). We have included these data as Figure 3 in the revised manuscript.

We have done actin-loading controls for the total lysates from all the replicates by re-probing membranes for actin. We are confident that our ponceau-S staining is sensitive enough to show equal loading, which is a standard practice in our lab and in the field. This allows us to look at total protein levels and not having to rely on a single protein.

Figure 5

● **As SENP6 depletion has considerable impact on cell cycle progression, is the increase in PML/DDR factor localisation due to an enrichment of a specific cell cycle stage? Do these co-localisations occur in all stages of the cell cycle?**

See point 1 on page 1 of this rebuttal.

Liebelt *et al.* (DOI: [10.1038/s41467-019-11773-x](https://doi.org/10.1038/s41467-019-11773-x)) show an increased amount of cells in G2/M and a concomitant decrease in cells in G1 with shRNA-mediated knockdown of SENP6 as determined by propidium iodide staining and flow cytometry; the number of cells in S-phase remained unchanged. Dou *et al.* (DOI: [10.1016/j.molcel.2010.07.021](https://doi.org/10.1016/j.molcel.2010.07.021)) show a delay in S phase and accumulation of cells in S and G2/M phases after cell synchronization in siSENP6 cells. In all our microscopy experiments, quantification was done on interphase cells, while mitotic cells were excluded. We therefore do not expect the increase in PML/DDR localization to be related to a relative increase in the number of cells in G2/M. At entry into mitosis, PML bodies are described to become de-SUMOylated and change their composition (reviewed in DOI: [10.3390/cells8080893](https://doi.org/10.3390/cells8080893)), so we can speculate that these colocalizations are unlikely to occur in mitotic cells.

● **Why do some factors co-localise with PML while others do not? What determines the difference?**

See point 1 on page 1 of this rebuttal.

We hypothesized that this is dependent on SUMO-SIM interactions and also addressed this with experiments in the manuscript. We included CENP-C as a control in our colocalization experiments, as the centromeric proteins do not contain functional SIMs and consequently cannot bind to SUMO *in vitro* (Liebelt *et al.* DOI: [10.1038/s41467-019-11773-x](https://doi.org/10.1038/s41467-019-11773-x)). To address the SUMO-SIM dependency mechanistically, we also included colocalization experiments with an ERCC1 SIM mutant and tri-SUMO fusion. We are aware that this localization might be further facilitated or is co-dependent on other factors (and not exclusively regulated by SUMO-SIM interactions) that can contribute to condensate formation, such as intrinsic disordered regions in proteins.

● **Most of the DDR factors analysed (BRCA1, 53BP1, RAP80, MDC1, ERCC1 and XPF) form DSB associated foci with exogenous DNA damage – do they do this in siSENP6 cells? Does the re-localisation to PML-NBs in undamaged cells prevent or reduce the relocalisation to DSBs?**

We would like to refer to our reply to comment #3, discussing our functional IR-induced DNA damage foci kinetics and UVA-laser induced DNA damage experiments.

Figure 6

- **Does SENP6 co-localisation with PML-NBs require SENP6 SIM motifs (characterised by Wagner et al 2019). Does SENP6 residency in the PML-NBs promote deSUMOylation and eviction of DDR factors, with loss of SENP6 causing retention in PML-NB bodies?**
- **What is the localisation pattern on SENP6 in the PML KO clones?**

See point 1 on page 1 of this rebuttal.

We do not describe SENP6 localization in PML bodies in our manuscript. Hattersley *et al.* (doi: [10.1091/mbc.E10-06-0504](https://doi.org/10.1091/mbc.E10-06-0504)) show SENP6 localization at PML bodies with overexpressed catalytically inactive GFP-SENP6, but not wildtype. We do not observe accumulation of SENP6 (endogenous or exogenously expressed wildtype and catalytically inactive GFP-SENP6) in body- or foci-like structures under our experimental conditions.

Reviewer #2 (Remarks to the Author):

Summary of manuscript

In this manuscript, Claessens and Vertegaal investigate the role of the poly-SUMO2/3 deconjugating enzyme SENP6 in genome stability. The authors suggest that the major function of SENP6 isn't directly at DNA lesions, but rather within the nucleoplasm, where it limits polySUMOylation of a number of factors involved in the DDR. They then suggest that the hyper-SUMOylation of a range of DDR factors doesn't play a major role in their ubiquitin-dependent degradation, as depletion of SENP6 leads to reduced polySUMOylation of some DDR factors. Instead, the authors claim that a number of polySUMOylated DDR factors, but not all factors examined, become enriched at PML bodies, which show a dramatic increase in number after SENP6 depletion. The authors suggest that the presence of SUMO-interacting motifs in DDR factors promotes their localisation to PML bodies, rather than merely their SUMOylation status, using ERCC1 as an example of this. Surprisingly, genetic abrogation of PML had no impact on the ability of various DDR factors to form foci after SENP6 depletion, with the authors suggesting that hyper SUMOylation of these factors is sufficient to promote formation of protein condensates. Overall, hyper polySUMOylation of DDR factors in the absence of SENP6 is proposed to drive SUMO-SIM interactions in both a PML-dependent and independent manner, which prevents recruitment to DNA lesions, impairing DNA repair and driving genome instability.

Strengths:

- **Convincing cellular phenotypes and cellular biochemistry, with data generally well controlled (see some exceptions below)**
- **Findings that will be of interest to a wide audience across DNA repair and ubiquitin/SUMO/UBL fields.**
- **Proposes a degradation-independent signalling function for SUMOylation in the regulation of DDR factors.**

Weaknesses:

- **Some data is repetitious with previous publications from the Vertegaal lab**
- **Lack of mechanism for foci formation after PML loss**
- **Suggested model is not entirely supported by the data**
- **Model is not universally applicable, yet no discussion is made as to why some DDR factors might be impacted and others not.**

- **SUMO-SIM model already a well known mechanism of PML body formation**

Overall:

Due to the major criticisms noted below, it would be difficult to support the manuscript in its current seven figure format, though, with additional controls and experiments to strengthen the current data, I could potentially support a shorter article format.

We thank the reviewer for support. We appreciate the extensive critical appraisal and suggestions for improvement and solidification of the proposed model.

Major points:

Point 1.

Figures 1-3 are essentially data that has already been published (except Fig 1C), so there's limited novelty here and the majority could go in the Supplementary. I'd suggest that these should be consolidated and the article made into a shorter format.

We appreciate the reviewer's suggestion to move some data to the supplement. The figures related to the mass-spectrometry data which has already been published are now included as Supplementary Figure 3 in the revised manuscript.

Point 2.

The suggestion that SENP6 functions within the nucleoplasm is based on the authors inability to detect SENP6 at IR-induced γ H2AX foci. However, given that DNA triggers PIAS-dependent SUMOylation of DDR factors (Psakhye and Jentsch, 2013), why would the deSUMOylation machinery act predominantly within the nucleoplasm as the authors claim? Furthermore, without performing similar localisation studies as previous studies, it seems rather pointless to make comparisons. Did the authors use biochemical fractionation for SENP6 localisation after DNA damage? Or did they try the PLA assay?

This comment is related to earlier comments from reviewer #1 raising similar points regarding SENP6 localisation and we would like to refer to our replies there. Our statement that SENP6 remains and functions exclusively in the nucleoplasm was phrased suboptimally. In the SENP6 localization experiment in the original manuscript, we failed to observe accumulation of SENP6 at IR-induced damage foci. However, with this, we did not mean to state that SENP6 is not present or does not function at the chromatin. Cellular fractionation assays in Wagner *et al.* (DOI: [10.1016/j.celrep.2019.08.106](https://doi.org/10.1016/j.celrep.2019.08.106)) and our own fractionation assays show SENP6 in the chromatin fraction in both the absence and presence of DNA damage in comparable amounts. In the immunofluorescence experiments that we did during the revision, we occasionally observed exogenously expressed GFP-SENP6 at UVA-laser induced DNA damage lesions. We have carefully rephrased the corresponding results section.

Point 3.

The authors state that "non-modified protein levels of CtIP, BLM and BRCA1 decreased, indicating that a buildup of SUMO2/3 polymers on these proteins could have led to their degradation or alternatively to such extensive SUMOylation that the amount of non-modified proteins was reduced". It is essential to provide more evidence to address this. To distinguish between the two, I suggest the authors take recombinant SENP6 and add it the cell lysate or pulldown, then blot back for CtIP, BLM, BRCA1 etc... At present, there's no reason proposed as to why polySUMOylation of these factors would be reduced after SENP6 depletion. A review from the Cohen lab explains how polyubiquitylation can lead to spurious conclusions derived from immunoblotting when the substrate is heavily modified by ubiquitin, which is analogous to the data presented here: <https://www.ncbi.nlm.nih.gov/pmc/articles/PMC4709362/>

Furthermore, the authors claim that "Taken together, these data rule out that SUMO2/3 polymers on these proteins are substrates for STUbLs and facilitate degradation." To test this more rigorously, the authors would have to assay protein turnover of these proteins over a timecourse.

See point 2 on page 1 of this rebuttal.

The reduction in polySUMOylation of the proteins tested in our experiment after SENP6 knockdown and MG132 treatment could be a result from depleting the free SUMO2/3 pool for SUMOylation of other substrates, like Myc, that become heavily SUMOylated upon MG132 treatment. To maximize the effect of proteasome inhibition, we used treatment times of 6-8 hours. Longer timepoints lead to excessive cell death and shorter timepoints might miss out on an effect.

Point 4.

A major weakness of the manuscript is the lack of mechanism for foci formation in the absence of PML. To start to understand the mechanism of foci formation in the absence of PML (Figure 7), the authors could at least use the SUMO E1i to see if SUMOylation is still required for foci formation of ERCC1, XPF, BRCA1, BLM and 53BP1 (similar to the experimental set-up in Figure 5). Presumably it will, so the authors can then use the ERCC1 WT vs SIM mutant understand if the SUMO-SIM model still applies in the absence of PML. In addition to this, it would be useful to characterise further the foci formed in the absence of PML and SENP6 – can the authors take a panel of known PML-associated factors and assess whether the PML-independent foci formation of the DDR factors still contains canonical PML-associated factors?

See point 1 on page 1 of this rebuttal.

We agree with the reviewer that this would be a valuable experiment and indeed expect the SUMO-E1 inhibitor to prevent foci formation/condensation in the absence of PML, like we show in parental U2OS. We also think it is interesting to further characterise the composition of these foci/condensates. We used the same experimental set-up in one of our PML knockout clones. Cells were mock-transfected (NTP) or with siSENP6 for 96 hours. Cells were additionally treated with ML792 for 24 hours prior to fixing. We did a co-staining for XPF and SP100, another major component of PML bodies, and single stainings for BRCA1, 53BP1 and BLM. Similar as in parental U2OS, we observed a significant increase in XPF foci in siSENP6 cells and this was prevented by ML792 treatment (A). There was also a significant increase in SP100 foci in siSENP6 cells, which was again rescued by additional ML792 treatment (A). We also observed colocalization of XPF foci with SP100 foci, suggesting that these condensates can contain other factors than PML (B-C). We observed similar phenotypes for BLM and 53BP1 as in parental U2OS, but not for BRCA1 (D). We have added these data to Supplementary Figure 10 in the manuscript.

See point 1 on page 1 of this rebuttal.

We are aware of this phenomenon and also discussed this as a possible explanation for why some PML bodies and damage foci appear to be juxtaposed: “However, in some cells, PML and γ H2AX occasionally appeared to be juxtaposed, which could potentially be explained by the observation that telomerase-negative cells contain PML bodies that associate specifically with chromatin at telomeres and are involved in alternative lengthening of telomeres (ALT; ALT-PML bodies).” We did

not further address this experimentally and agree that this would be valuable to determine. Li *et al.* (DOI: [10.1038/s41467-023-37480-2](https://doi.org/10.1038/s41467-023-37480-2)) very recently described the requirement of SENP6-mediated de-SUMOylation for telomere de-clustering following ALT-directed telomere synthesis, preventing chromosome mis-segregation and mitotic cell death.

Point 6.

The authors propose in their model that the hyper polySUMOylation of DDR factors leads to their accumulation in PML and SUMO condensates, which inhibits their ability to localise to DNA damage sites. However, there is no data in the manuscript which shows that the DDR factors described fail to properly localise to genotoxin-induced DNA lesions in the absence of SENP6 – quantification +/- DNA damage would be required.

We have addressed this in comment #3 from reviewer #1 and would like to refer to our reply there.

Point 7.

Why is median number of foci often used throughout the manuscript?

As is visible from the number of foci per individual nucleus analysed, plotted in addition to the average number of foci per nucleus per biological replicate, the distribution of the number of foci across nuclei does not follow a normal distribution. Therefore, summarizing/averaging the data per biological replicate with the median is statistically correct. We also applied this to other quantifications where the data was not normally distributed (e.g. % of DDR⁺ PML bodies). For normally distributed data, the mean was used.

Minor points:

Figure 1:

• B - SUMO1 would be a useful control here

We did a γH2AX and SUMO1 co-staining in mock-transfected (NTP) and siSENP6-transfected cells. There is a minor, although significant, increase in the number of cells with more than 5 colocalizing foci. However, this percentage is much lower than for SUMO2/3 (average of 5% for SUMO1 and 20% for SUMO2/3 across three biological replicates, the latter shown in the Figure 1D in the manuscript). We have included these data as Supplementary Figure 1C in the revised manuscript.

We have removed this figure from the manuscript.

● **F - It is not clear how many cells were quantified to assess micronuclei formation following siSENP6.**

Micronuclei were quantified using the same microscopy data as in Figure B-E: “Left panel: representative images of cells in (B) with micronuclei. Scale bar indicates 10 μ m. Right panel: average percentage of cells in (B) with ≥ 1 micronuclei and with ≥ 1 γ H2AX⁺ micronuclei. Data are shown as mean \pm SD from three biological replicates. Significance was determined with an unpaired t-test.” Therefore, we only mentioned the numbers of cells once when first mentioning the quantification of the data.

● **G - SENP6 depletion is lethal, as illustrated by DepMap and the authors own previous clonogenic data. Thus, presumably there is still some level of SENP6 in the cells after 5 days. This caveat should be stated in the text.**

The reviewer is right that there is still some level of SENP6 in the cells after 5 days. We previously evaluated SENP6 levels throughout the assay. However, the expression is almost entirely gone 24h post-transfection, when re-seeding for DNA damage treatments the next day. We believe that the expression 5 days post-treatment (7 days post-transfection), is because we do not do repetitive siRNA transfections throughout the assay to keep the levels depleted, so cells will eventually start re-expressing the protein.

We agree with the reviewer that performing rescue experiments would be a nice addition. However, we would like to argue that there is a sufficient amount of literature available on SENP6 protease activity and its function in removing SUMO2/3 chains from substrates. Therefore, we did not prioritize this experiment.

A SUMO1 blot should also be included here.

Hattersley *et al.* (doi: [10.1091/mbc.E10-06-0504](https://doi.org/10.1091/mbc.E10-06-0504)) already showed on total lysates from HeLa cells that SENP6 knockdown predominantly leads to an increase in SUMO2/3 conjugates, rather than SUMO1. Moreover, Shick *et al.* (DOI: [10.1038/s41467-021-27704-8](https://doi.org/10.1038/s41467-021-27704-8)) showed in total lysates from lymphoma cells that low levels of SENP6 mainly lead to an increase in SUMO2/3 conjugates and not SUMO1 conjugates. We confirmed these observations for three biological replicates with shRNA mediated knockdown of SENP6. We have included these data as Supplementary Figure 4A in the revised manuscript.

Figure 4:

- **It would be better to place the inputs and pulldown for each target together, rather than split them across different panels.**

See point 2 on page 1 of this rebuttal.

Figure 5:

- **It would be important to split the green and purple channels, as well as having the merged image.**

We have now included the split channels, besides the merged image.

Figure 6:

- **The ERCC1-GFP(SIM) mutant seems to be much more cytoplasmic than WT – a biochemical fractionation would illustrate its distribution. Is there sufficient nuclear localisation to assess its recruitment to foci?**

We were indeed aware of the partial cytoplasmic localization of ERCC1-GFP_{SIM}. At the time, we questioned whether there was increased nuclear export of the protein, perhaps through CRM1, but a pilot test with CRM1 inhibition did not prevent cytoplasmic localization. We have now quantified the amount of ERCC1-GFP_{WT/SIM} in the mock-transfected cells (NTP) from our microscopy experiment to assess nuclear expression levels. GFP fluorescence intensity was quantified by measuring the intensity in the nuclei and subtracting this with the fluorescence intensity measured in the background of each image. Cells from three biological replicates were pooled and plotted. Error bars represent mean±SD from the pooled cells. There was a large spread in GFP nuclear intensity for both cell lines and the average intensity was actually slightly higher in the ERCC1-GFP_{SIM} cell line, despite the cytoplasmic localization (significance was determined with an unpaired t-test). We therefore believe that the amount of ERCC1-GFP_{SIM} is sufficient for recruitment to foci. We also noted slightly higher overall expression levels of ERCC1-GFP_{SIM} by western blotting, which might result from a combination of slightly higher nuclear levels and cytoplasmic expression.

- **Are the SUMOylation sites of ERCC1 well defined? It might be useful to introduce K>R mutants to complement the SIM mutant data.**

The SUMOylation sites of ERCC1 are not defined and ERCC1 has 19 lysine residues which could be potential sites. We do not have good experience with making K>R mutants in a short amount of time, as we often observe hopping of SUMO to other lysines. We do not think that this is a worthwhile endeavour for the current manuscript, and would be more suited for a paper focusing entirely on ERCC1 SUMOylation.

- **D - Running the input and IP samples on the same gel would enable improved comparison to the input levels.**

All samples shown were run on the same gel, but we chose different exposures for visual purposes to prevent overexposure of the input samples. Here, we are mainly comparing the amount of bound ERCC1 in the elutions between wildtype ERCC1 and the potential SIM mutants.

- **H - The immunoblots for the quantification of co-purified XPF could be shown in parallel.**

The immunoblots of one replicate are already included in the supplement (Supplementary Figure 8C).

Figure 7:

- **A - Higher magnification images may be useful to demonstrate the localisation of PML and γH2AX foci in response to SENP6 knockdown.**

We imaged with a 63x objective. We included zoomed-in inserts to better demonstrate the localization of some PML and γH2AX foci.

Reviewer #3 (Remarks to the Author):

The manuscript by Claessens and Vertegaal extends previous work by the Vertegaal laboratory (Mitra et al., Nature communications, 2019) and other research groups (Gibbs-Seymour et al., Mol Cell, 2015, Dou et al., Mol Cell, 2010, Wagner et al., Cell reports 2019) on the importance of SENP6 for the maintenance of genomic stability. As such, the manuscript is clearly written, the experiments are of high quality and data are

presented with proper controls. However, some results (large parts of Figure 1 and Figure 2) are just a reiteration of already available/published data and therefore provide little novelty. Along the same line, Figure 3 is solely a validation of the previously published mass spectrometry (Mitra et al., Nature communications, 2019). Furthermore, it was previously described that SENP6 depletion leads to retention of SUMO chains on a subset of DDR factors leading to reduced chromatin recruitment (Schick et al., 2022). The most interesting novel aspect provided here is the observation that at least for a subset of repair factors this is due to their hyperaccumulation in nuclear condensates, such as PML speckles. This is in line with the described role of SENP6 as a regulator of PML body (Hatterslay et al., MCB 2011, Mukhopaday et al., JCB, 2006) and strengthens the concept that SUMO-SIM interactions act as a scaffold of PML body and other nuclear SUMO condensates (Banani et al., Cell, 2016). Altogether, the manuscript provides a mechanistic explanation how knock-down of SENP6 reduces the presence of DDR factors at sites of DNA damage. The authors propose that supraphysiological levels of polySUMO chain induce the trapping of DDR factors in nuclear condensates. While this is a potentially interesting finding the conceptual novelty of the work is somewhat limited. Further, some aspects of the work need to be clarified or strengthened.

We thank the reviewer for supporting our manuscript. We appreciate the extensive critical appraisal and suggestions for improvement and solidification of the proposed model.

Major points:

1) Data presented here support the idea that polySUMO chains exhibit important regulatory functions beyond their function as recruitment platforms for the SUMO targeted ubiquitin ligase RNF4. However, I do feel that this aspects needs to be solidified, because previous work has shown that polySUMOylation of DDR factors result in their interaction with RNF4 and subsequent degradation after DNA damage (for example Galanty et al., Genes and Dev, 2012, Garvin et al., 2019). Further, RNF4 is known to reside at PML body possibly indication that trapping in condensates precedes degradation. Of note, previous work has shown that ablating the expression of SENP6 also leads to ubiquitin-mediated proteolysis of RNF4. Thus, it is conceivable that the DDR factors investigated here might represent StUbL substrates under physiological conditions (e.g. regulated acute inactivation of SENP6), but are not degraded in the experimental system used here due to the constitutive inactivation of SENP6 followed by loss of RNF4. Therefore, RNF4 levels in SENP6 depleted cells should be controlled. Further, the fate of the polySUMOylated DDR factors upon re-expression of RNF4 should be monitored. To circumvent the drawback of constitutive SENP6 inactivation by siRNA the authors may also consider induced degradation of SENP6 protein, which may better reflect the acute regulation of SENP6 (Dou et al., Mol Cell, 2010).

We thank the reviewer for pointing out this potential caveat. Indeed Rojas-Fernandez *et al.* (DOI: [10.1016/j.molcel.2014.02.031](https://doi.org/10.1016/j.molcel.2014.02.031)) have described that the accumulation of SUMO chains in the absence of SENP6 induces RNF4 autoubiquitination and degradation.

We determined RNF4 levels on total lysates from two biological replicates of our His10-SUMO2 pulldown experiments with MG132 treatment and did not observe degradation of RNF4 for shRNA SENP6#1 and slightly lower levels for shRNA SENP6#2, so we are confident that the build-up of SUMO2/3 chains on the DNA repair proteins in this experimental setting is exclusively a result of the absence of SENP6. We have included these data as Supplementary Figure 4C in the revised manuscript.

We also determined RNF4 levels on total lysates from an siRNA knockdown experiment where we lysed cells 72h and an experiment where we lysed cells 96h post-transfection, and observed an almost complete loss of RNF4 expression. Treatment with 1 μ M SUMO-E1 inhibitor ML792 for 24h rescued these levels. We agree with the reviewer that we cannot exclude a potential combinatorial effect of SENP6 and RNF4 depletion in our experiments done with siSENP6-treated cells and added this remark to the revised manuscript. Representative blots of two biological replicates are shown. We have included these data as Supplementary Figure 5A in the revised manuscript.

We followed the reviewer's suggestion to re-express RNF4 in the siSENP6 knockdown cells. We attempted to make a stable Flag-RNF4 cell line with a lentivirus strategy, to repeat our siSENP6 experiments, but all cells died after the exogenous Flag-RNF4 expression. We expect this to be from too high total RNF4 levels, causing cellular toxicity. We then tried to transduce cells with Flag-RNF4 after siSENP6 knockdown, when RNF4 levels start to drop, to avoid RNF4 overexpression from the exogenous Flag-RNF4 and the endogenous RNF4. We initially transduced cells 24h post siRNA transfection and processed the cells 48h post-transduction (and 72h post-transfection). This resulted in less massive cell death, but extremely high RNF4 levels (A). We then attempted different timings of the Flag-RNF4 transduction: we transduced cells 24h, 48h or 72h after siSENP6 transfection, then processed the cells after 48h, 24h or 8h post-transduction, respectively (all 72h post siRNA transfection). We also lowered the MOI to 1 and 0.5 (instead of our standard MOI of 3). These conditions resulted in less cell death and lower Flag-RNF4 levels on total lysate level by western blotting (B). However, Flag and γ H2AX immunofluorescence showed that lower total RNF4 levels resulted from lower percentages of cells being transduced/expressing Flag-RNF4 rather than lower Flag-RNF4 levels per cell; cells expressing Flag-RNF4 expressed it at a high level (C). Since RNF4 is a critical protein in the DNA damage response and we observed high toxicity from the expression of Flag-RNF4 in parental U2OS, we cannot use these high RNF4-expressing siSENP6 cells to determine a rescue phenotype. We do not think attempting transient transfection of exogenous RNF4 is worthwhile, because overexpression artefacts tend to be an even bigger problem.

We also attempted other approaches in parallel to get a better idea of a role for the SENP6-RNF4 axis in regulating the SUMO2/3 levels of our identified DNA repair proteins. We performed shRNA-mediated knockdown of SENP6, RNF4 and both in our His10-SUMO2 pull-down experiment to determine whether co-depletion of RNF4 leads to further build-up of SUMO2/3 chains compared to SENP6 depletion alone, which could point towards recognition of the SUMOylated proteins by RNF4, followed by their degradation. We blotted for SUMO2/3, BARD1, BRCA1, BLM and MDC1. SUMO2/3 levels were further increased with the double knockdown compared to SENP6 alone, suggesting that there are proteins becoming increasingly SUMOylated. For BRCA1, BARD1 and BLM, double knockdown of RNF4 and SENP6 led to a further increase in SUMOylation levels compared to SENP6 knockdown alone; for BRCA1 and BLM this was already visible on total lysates (A). We quantified the changes in SUMOylation levels of the DNA repair proteins for two biological replicates by dividing the intensity in shRNA SENP6#1 + RNF4#1 and shRNA SENP6#1 + RNF4#2 cells with the intensity in shRNA SENP6#1 cells (B). We cannot formally exclude that RNF4 knockdown indirectly causes these changes, as SUMO E3 ligases have been identified as RNF4 targets as well. Therefore, we cannot conclude from just this experiment that these proteins are direct RNF4 targets. Nonetheless, it does illustrate that in cells depleted of both proteins at least a subset of DNA repair proteins becomes more excessively SUMOylated than in the absence of SENP6 alone and that we are potentially looking at consequences of supraphysiological levels of repair protein SUMOylation that perhaps do not occur under physiological conditions during DNA damage signalling. This could potentially be interesting in the context of tumours where SENP6 levels are consistently dysregulated, like described in Shick *et al.* (DOI: [10.1038/s41467-021-27704-8](https://doi.org/10.1038/s41467-021-27704-8)). We have included these data as Figure 5 in the revised manuscript.

A

2) The authors conclude that, in the absence of SENP6, supraphysiological levels of polySUMO chains induce the trapping of DDR factors in nuclear condensates thereby

reducing their presence at sites of DNA damage. At this stage, however, the physiological relevance and regulation of this process remains elusive.

This comment is related comment #3 of reviewer #1.

Is the association of SENP6 with the respective DDR factors induced upon DNA damage in order to trigger their recruitment to sites of DNA damage?

This comment is related to comment #2 of reviewer #1.

Is the activity of SENP6 altered in response to genotoxic insults?

To address this we performed pilot tests with a Rho-SUMO2-PA probe and cell lysates from cells treated with different genotoxic insults (HU, CPT and MMC), but did not observe any consistent or reproducible changes in activity. Garvin *et al.* (doi: <https://doi.org/10.1101/2022.03.23.485504>) also assessed SENP6 activity after IR treatment with a SUMO2-VS probe and did not detect any differences in activity, while they did for SENP1. We would like to speculate about possible scenarios. We could hypothesize that SENP6 is constitutively active and its activity is not altered under the tested conditions. In that situation, SENP6 functions to continuously maintain proteins in a hypo-SUMOylated state and perhaps it is rather timely subcellular localization or physical association with its substrates that regulates their deconjugation by SENP6 than differences in SENP6 activity. However, we cannot rule out that the experimental system and conditions used here might be limited to reliably assess changes in SENP6 activity, as these could be very subtle, local and/or transient changes.

Besides SENP7, the other SENP proteins do not specifically remove SUMO2/3 chains from proteins. We therefore deem it unlikely that depletion of one of these SENPs would lead to condensate formation of the DDR proteins identified in our SENP6 screen. We do not rule out that these DDR proteins could also be potential substrates of SENP7, the other SUMO-chain deconjugating enzyme, but we believe that this is beyond the scope of the current manuscript, as this warrants thorough investigation of SENP7 substrate identification and potential regulation of the DNA damage response. A potential (partial) overlapping role for SENP7 also would not weaken the involvement of SENP6. In siSENP6 cells, there is spontaneous induction of γ H2AX foci. We checked if knockdown with a pool of SENP7 siRNAs (siSENP7) can also induce this phenotype, but did not observe this (A). Knockdown of SENP6 and SENP7 was confirmed by immunoblotting (B). We have added these data as Supplementary Figure 1D in the revised manuscript.

With regards to mono- vs polySUMOylation, we have now also included pulldown experiments with shRNA-mediated SENP6 knockdown in our His10-SUMO2 wildtype and His10-SUMO2 K0 (lysine-less) cell lines and performed western blotting against a selection of our identified SENP6 targets. We predominantly observe polySUMOylation of these proteins in the absence of SENP6 rather than multi-mono-SUMOylation (data is shown and discussed in more detail in a comment from reviewer #1: Figure 3, third bullet point). **Additional points:**

3. In Figure 3, the input lanes for BLM and CtIP seems to have less respective protein levels in response to SENP6 depletion. At least for BLM, higher molecular weight regions also did not show any smear/bands. So, the claim that the decrease of the protein levels of BLM and CtIP can partially be because of reduction of SUMO-unmodified forms of the proteins is not justifiable.

See point 2 on page 1 of this rebuttal.

5. The chain specificity of Ub (K48, K63) in Figure 4D is worth exploring in the current context.

See point 2 on page 1 of this rebuttal.

In this experiment, we blotted for Ub K48 and K63 on two biological replicates of our His10-SUMO2 pulldown samples. Knockdown of SENP6 led to an increase in both K48 and K63 Ub chains on the His10-SUMOylated proteins. Knockdown of SENP6 combined with MG132 treatment did not cause an additional increase in K48 chains (shRNA SENP6#2 alone in one of the replicates did not give the expected phenotype). There appeared to be a minor increase in K63 chains compared to SENP6 knockdown alone. We do believe that one should be cautious of potential exhaustion of the system and free Ub in the SENP6 knockdown cells treated with MG132.

6. Figure 6C, D: ERCC1 SIM mutants seem to have more expression/stability in cells. Can the authors provide some explanations for this?

We thank the reviewer for pointing this out and noted this as well at the time of making the cell lines. We could speculate that the ability of the protein to bind to SUMO affects the turnover or stability of the protein, but we have not done experiments addressing this. Rather than being a functional difference, it could also be a consequence of a technical aspect in the retroviral making of the cell lines. Since we observe cytoplasmic expression for ERCC1-GFP_{SIM} and not for the wildtype protein, and also slightly higher nuclear levels, the higher overall expression could be a result of this (data is shown in a related comment from reviewer #2: Figure 6).

7. As such, Figure 6H is not relevant to the observations presented here and can go in the supplementary section. ERCC1 SIM domain rather than ERCC1 SUMOylation seems to be important for localizing to PML bodies and for interaction with XPF. However, ML792 treatment sensitizes ERCC1-XPF1 interaction. Does SUMOylated XPF1 binds to ERCC1 by the SIM domain of the latter?

We disagree that this observation was not relevant in the original manuscript. It is not solely the SIM domain that is important for localization to PML bodies, as we show that ERCC1 localization to PML bodies is induced in the absence of SENP6, when polySUMO chains build up on the protein, and does not happen under control conditions where the protein is not polySUMOylated, or with additional ML792 treatment, counteracting the build-up of polySUMO chains induced by SENP6 depletion by blocking SUMO conjugation. However, we also provide evidence that this is not sufficient and that SIM domains are likely also required, so it being a combination of covalent SUMOylation and non-covalent SUMO binding that enhances this localization. In the case of

ERCC1, it seems that heterodimer formation with XPF through the HnH2 domain is somehow also SUMO-SIM dependent. We think the speculation of the reviewer that SUMOylated XPF might bind the SIM domain of ERCC1 is very interesting. However, we did not follow up on this as our manuscript is not primarily focused on the role of SUMO in ERCC1-XPF endonuclease function and believe that this would warrant proper investigation.

8. In PML KO cell line, do foci for all these DDR proteins co-localize? As ERCC1-XPF is shown to be SUMO-SIM dependent, at least these two proteins should co-localize. Are any non-canonical foci formed in absence of PML where some of these polySUMOylated proteins co-localize? If so, does ML792 treatment dissolve these foci (not shown in Figure 5D)?

See point 1 on page 1 of this rebuttal.

This is related to a comment from reviewer #2 (Point 4) also asking for further characterization of the PML and PML-independent condensates and would like to refer to our reply there.

Reviewers' Comments:

Reviewer #1:

Remarks to the Author:

The authors have answered my comments sufficiently. The new mechanism proposed is better supported by the data provided, and I am happy to recommend publication in Nature Communications.

Reviewer #2:

Remarks to the Author:

I thank the authors for their careful responses to the criticisms, comments, and questions raised by all the reviewers. The authors have made a great effort with their revised manuscript to address most of the criticisms, comments, and questions from all three reviewers to revise and clarify their model, either through further experimental work or through well-reasoned and honest responses. Though I'm ambivalent on this, the manuscript is boarding on being unwieldy with its nine figures, so the authors might consider moving some to the Supplemental Figures. Regardless, I support its revised form.

Reviewer #3:

Remarks to the Author:

Cleassens and co-workers now provide a significantly revised version of their initial work. The authors undertook a considerable effort to address the main issues I raised on the initial version. The new data clarify some concerns, such as the involvement of RNF4 in the proposed model (My major point #1). Further, the authors addressed the physiological relevance of condensate formation upon depletion of SENP6 and the potential functional implication of PML in this process (My major points #2 and #8). Based on new data the authors now propose a revised version of their initial model. This revised model predicts that depletion of SENP6 leads to supraphysiological levels of SUMOylation, which activates the RNF4 pathway resulting in uncoordinated recruitment and persistence of SUMOylated proteins at DNA damage sites, as well as their accumulation in (PML) nuclear condensates.

I do appreciate the experimental efforts of the authors to solidify their conclusions. I also acknowledge the openness to revise their initial model. However, I do feel that at its current stage the work largely remains at a descriptive level and provides only limited novelty with respect to published data. Parts of the manuscript is a validation of their published MS data (Liebelt et al., 2019). Further, several papers have connected SENP6 to DNA repair and the localization of DDR proteins at chromatin (Dou et al., Mol Cell 2010; Gibbs-Seymour et al., 2015; Garvin et al., Genes Dev., 2019, Wagner et al., 2019). Also, some aspects added in the revised version, such as SENP6's role in limiting polySUMOylation of DDR factors in response to HU, have been previously addressed on selected substrates, such as BLM (Ellis et al., 2021). What is missing, at least to my opinion, is a conclusive mechanistic explanation how SUMO polymers dictate the location of DNA damage factors. The initial observation that the lack of SENP6 impairs a proper DDR response by trapping DDR factors in nuclear condensates was very attractive, but this aspect could not be solidified by experimental data. With respect to the revised model, it remains somewhat counterintuitive how the trapping of DDR factors, such as BRCA1, in nuclear condensates, is at the same time compatible with their persistence at DNA damage. It would be important to explore whether this indeed occurs "in parallel" (as proposed by the authors) or in a sequential manner and whether and how the trafficking of selected DDR repair factors, such as BLM or BRCA1, between DNA damage sites and condensates is controlled by the interplay of SENP6 and RNF4.

Altogether, despite the wealth of data and several interesting aspects, I feel that the manuscript does

not provide very substantial novelty and mechanistic insight. Therefore, I am somewhat hesitant to support its publication in Nature communications.

REPLY TO REVIEWERS' COMMENTS

Reviewer #1 (Remarks to the Author):

The authors have answered my comments sufficiently. The new mechanism proposed is better supported by the data provided, and I am happy to recommend publication in Nature Communications.

We thank the reviewer for their critical appraisal and useful suggestions to improve on our manuscript during the revisions, and for their support of its revised form.

Reviewer #2 (Remarks to the Author):

I thank the authors for their careful responses to the criticisms, comments, and questions raised by all the reviewers. The authors have made a great effort with their revised manuscript to address most of the criticisms, comments, and questions from all three reviewers to revise and clarify their model, either through further experimental work or through well-reasoned and honest responses. Though I'm ambivalent on this, the manuscript is boarding on being unwieldy with its nine figures, so the authors might consider moving some to the Supplemental Figures. Regardless, I support its revised form.

We thank the reviewer for their critical appraisal and useful suggestions to improve on our manuscript during the revisions, and for their support of its revised form. We have moved panel 8c and 8f to the supplement. Panel 9f is now separately included as Figure 10 on request of the editor.

Reviewer #3 (Remarks to the Author):

Cleassens and co-workers now provide a significantly revised version of their initial work. The authors undertook a considerable effort to address the main issues I raised on the initial version. The new data clarify some concerns, such as the involvement of RNF4 in the proposed model (My major point #1). Further, the authors addressed the physiological relevance of condensate formation upon depletion of SENP6 and the potential functional implication of PML in this process (My major points #2 and #8). Based on new data the authors now propose a revised version of their initial model. This revised model predicts that depletion of SENP6 leads to supraphysiological levels of SUMOylation, which activates the RNF4 pathway resulting in uncoordinated recruitment and persistence of SUMOylated proteins at DNA damage sites, as well as their accumulation in (PML) nuclear condensates.

I do appreciate the experimental efforts of the authors to solidify their conclusions. I also acknowledge the openness to revise their initial model. However, I do feel that at its current stage the work largely remains at a descriptive level and provides only limited novelty with respect to published data. Parts of the manuscript is a validation of their published MS data (Liebelt et al., 2019). Further, several papers have connected SENP6 to DNA repair and the localization of DDR proteins at chromatin (Dou et al., Mol Cell 2010; Gibbs-Seymour et al., 2015; Garvin et al., Genes Dev., 2019, Wagner et al., 2019). Also, some aspects added in the revised version, such as SENP6's role in limiting polySUMOylation of DDR factors in response to HU, have been previously

addressed on selected substrates, such as BLM (Ellis et al., 2021). What is missing, at least to my opinion, is a conclusive mechanistic explanation how SUMO polymers dictate the location of DNA damage factors. The initial observation that the lack of SENP6 impairs a proper DDR response by trapping DDR factors in nuclear condensates was very attractive, but this aspect could not be solidified by experimental data. With respect to the revised model, it remains somewhat counterintuitive how the trapping of DDR factors, such as BRCA1, in nuclear condensates, is at the same time compatible with their persistence at DNA damage. It would be important to explore whether this indeed occurs "in parallel" (as proposed by the authors) or in a sequential manner and whether and how the trafficking of selected DDR repair factors, such as BLM or BRCA1, between DNA damage sites and condensates is controlled by the interplay of SENP6 and RNF4.

Altogether, despite the wealth of data and several interesting aspects, I felt that the manuscript does not provide very substantial novelty and mechanistic insight. Therefore, I am somewhat hesitant to support its publication in Nature communications.

We thank the reviewer for their critical appraisal and useful suggestions to improve on our manuscript during the revisions. We understand the points raised above regarding our original and revised model. We agree that there are limitations to our work and that there are still outstanding questions that need be addressed to fully understand the complex regulation of DNA damage response proteins by SENP6, the interplay with RNF4 and if and how persistence of proteins at sites of DNA damage and in nuclear condensates are linked. We have rephrased "in parallel" throughout our manuscript to remove any premature speculation or conclusions on this matter. Although there are now several studies that have linked SENP6 to the DNA damage response, more research is necessary to fully understand the different modes of regulation by SUMO and SENP6 for the functioning of distinct DNA damage response proteins under different physiological conditions. We have now also highlighted limitations and remaining open questions in the discussion.